# Robust Algorithmic Recourse Design Under Model Shifts

## Abstract

Algorithmic recourse offers users recommendations for actions that can help alter unfavorable outcomes in practical decision-making systems. Although many methods have been proposed to design easily implementable recourses, model updates or shifts may render previously generated recourses invalid. This paper addresses this challenge with two key contributions: 1) We introduce an uncertainty quantification method that calculates a theoretical upper-bound for the recourse invalidation rate. 2) We introduce a framework that allows users to manage the trade-off between the implementation cost of recourses and their robustness. This framework leverages the proposed invalidation rate bounds to generate recourses, catering to user-defined robustness requirements. Numerical results on multiple datasets demonstrate the effectiveness of the derived theoretical bounds and the efficacy of the proposed algorithms.

## 1 Introduction

Predictive models are being increasingly deployed in diverse consequential decision-making applications across a variety of contexts, such as loan applications (Moscato et al., 2021), job applications (Schumann et al., 2020), and criminal justice (Brayne & Christin, 2021) etc. Consequently, designing models that can provide individuals with explanations and recommendations to change their situation favorably is crucial, and even considered a legal necessity (Voigt & Von dem Bussche, 2017). For example, consider that there is an individual seeking a mortgage to purchase a home and the loan-granting institution uses a binary classifier and denies the loan application based on his/her attributes. It is important to provide recommendation of actions so that the user can improve his/her chance of loan being approved next time. Motivated by this, there are many interesting methods proposed to generate recourses for adversely affected individuals that are easy for users to implement (Wachter et al., 2017; Ustun et al., 2019; Karimi et al., 2020a). For instance, Wachter et al. (2017) propose a gradient-based approach to find the closest counterfactual to any negatively-predicted sample that results in a favorable prediction result. Ustun et al. (2019) propose an efficient integer programming approach to generate a list of actionable recourses for linear classifiers. Karimi et al. (2020b; 2021) have taken the causal relationships between features into account to investigate the nearest recourse from the perspective of minimal intervention.

These approaches work well if the underlying predictive models do not change. However, in practice, underlying predictive models often change for various reasons. For example, in the context of loan application, the underlying distribution of the population seeking loans may shift over time due to changes in economic conditions or shifts in consumer behavior. Moreover, as the data distribution shifts over time, some decision making institutions regularly retrain their models and/or use online learning frameworks to continuously adapt to new patterns in the data (as shown in Figure 1). This raises concerns about robustness and reliability of the generated recourse as previously prescribed recourse may no longer be valid once the model is updated. Recent studies have demonstrated the vulnerability of the generated recourse to minor distribution shifts or small model changes (Rawal et al., 2021). To address this issue, one strategy, coming from adversarial robustness and robust optimization literature, is to apply the classic minimax objective functions to design robust recourses that can withstand distributional perturbations or model changes (Upadhyay et al., 2021). These approaches use gradient-based methods to solve proposed minimax problems. Nevertheless, in order to generate a recourse that always leads to a positive outcome, the recourse cost is usually higher

and it is more difficult for the user to implement such recourse. Specifically, it has been observed that the recourse can only achieve one of low cost or robustness to model shifts (Pawelczyk et al., 2022).

To design recourses that are robust to model shifts while maintaining lower costs, there are two main challenges that need to be addressed: 1) determining how to measure the robustness of a given recourse under an unknown shifted model; and 2) devising a method to generate a model-agnostic recourse with the smallest possible cost while satisfying the user's robustness constraint. To tackle these challenges, we introduce the concept of the recourse invalidation rate under model shift, which represents the conditional probability that a given recourse becomes invalid under a potential shifted model. To bound the invalidation rate, we utilize conformal predictive inference techniques (Romano et al., 2019; Tibshirani et al., 2019) that can produce prediction set (almost exactly) for any given coverage level. In the context of algorithmic recourse, instead of setting a coverage level, we treat the coverage parameter as a variable and obtain a probability inequality for any given recourse, which will help measure the recourse robustness.

The second challenge in generating model-agnostic recourses lies in accommodating users with different levels of tolerance for recourse invalidation under model shifts. While some users may prioritize low invalidation rates, others may be willing to accept a higher rate in exchange for lower implementation costs. Existing approaches fail to provide users with the ability to navigate the trade-off between cost and robustness. To address this issue, we leverage our bounds on the recourse invalidation rate to produce recourses tailored to user's needs. In particular, we consider cases where users desire recourses with specific invalidation rates (as shown in Figure. 2). To design algorithms that guide recourse search towards regions of low cost while satisfying the invalidation rate constraint, we formulate a constrained optimization problem. However, the constraint of the proposed optimization problem is simulation-based and traditional optimization methods cannot be applied directly. To address such difficulty, we propose an extended alternating direction method of multipliers (ADMM) approach to efficiently find the minimal cost recourse that satisfies the invalidation rate constraint.

## 2 Background and Related Work

In this section, we provide related work on robust recourse as well as the user-involved recourse design. Background information regarding algorithm recourse and conformal predictive inference that are relevant to this work can be found in Appendix A.

**Robust recourse:** The robustness of algorithm recourse/counterfactual explanation has been studied in different settings, such as adversarial robustness, robustness to distributional shifts or model shifts. Adversarial robustness considers small uncertainty in the features of users and it has been shown that recourse methods that aim to offer recourses with minimal costs may not be adversarially robust (Dominguez-Olmedo et al., 2022). Then different methods have been proposed to generate counterfactual explanations with lower cost under a slight perturbation (Slack et al., 2021a). For the robustness under distributional shifts, Rawal et al. (2021) make the first attempt to understand how model updates resulting from data distribution shifts impact the recourses generated by state-of-the-art algorithms based on empirical analysis. Pawelczyk et al. (2020) study counterfactual explanations under predictive multiplicity, followed by (Black et al., 2021) which shows that recourses can even be invalid for models retrained with different initial conditions. To address these challenges, Upadhyay et al. (2021) propose a novel framework, Robust Algorithmic Recourse (ROAR), that leverages adversarial training for finding recourses that are robust to model shifts. However, ROAR's applicability is limited to linear classifiers. For nonlinear models, it approximates them with a linear model using LIME (Ribeiro et al., 2016), with robustness represented by perturbations of the parameters of the linear surrogate. Recent research has highlighted some limitations of the locally linear model of LIME, including fidelity (White & d'Avila Garcez, 2020) and robustness (Slack et al., 2021b). Similarly, Bui et al. (2022) also focus on linear classification settings and proposes the Counterfactual Plan under Ambiguity (COPA) framework, which constructs a counterfactual plan by minimizing a weighted sum to account for the model's uncertainty. However, the limitation of these methods to linear models or linear approximations is apparent, and several approaches have been proposed to address this limitation. For instance, Nguyen et al. (2022) propose a method to generate Bayesian recourse that is robust to data perturbations in a Gaussian mixture ambiguity set prescribed by the optimal transport distance. Additionally, Nguyen et al. (2023) present the

Distributionally Robust Recourse Action (DiRRAc) framework, which formulates a min-max optimization problem to generate recourse actions that have a high probability of being valid under a mixture of model shifts.

**User-involved recourse design:** User-involved recourse design refers to the process of incorporating user preferences or feedback when generating recourses. A recent paper (Pawelczyk et al., 2022) addresses the problem of enabling users to balance the trade-offs between the cost of generating recourses and their robustness to noisy implementations. Unlike our paper, which focuses on the robustness of recourses to model shifts, it investigates the robustness of recourses to noisy human responses. Specifically, the model allows users to select the probability of recourse invalidation when implementing the recourse with some level of noise. Under this setting, the paper proposes a measure of recourse invalidation based on the expected difference between the prediction value of the prescribed recourses and their implemented counterparts. The authors demonstrate that generating robust recourses in the face of noisy human response requires an additional cost. Furthermore, assuming that the distribution of the human response noise is Gaussian, Pawelczyk et al. (2022) provide a theoretical upper-bound for the proposed measure of recourse invalidation using a first-order approximation. De Toni et al. (2023) propose a human-in-the-loop approach for generating recourses using preference elicitation. The authors use a polynomial procedure to maximize the expected utility of selection and refine cost estimates iteratively in a Bayesian fashion. De Toni et al. (2023) demonstrate empirically that it can lower intervention costs with a handful of queries compared to user-independent alternatives.

## 3 Problem Formulation

Consider a binary classification problem of predicting labels $y \in \mathcal{Y} = \{0, 1\}$ from features $\boldsymbol{x} \in \mathcal{X} \subset \mathbb{R}^d$. Here, 0 is the unfavorable outcome and 1 is the favorable outcome. Given a dataset $\mathcal{D}$, we randomly split it into a training fold $\mathcal{D}_{\text{train}}$ and a calibration fold $\mathcal{D}_{\text{calib}} = \{(\boldsymbol{X}_i, Y_i)\}_{i=1}^n$. From the training fold $\mathcal{D}_{\text{train}}$, we derive the current ($t = 0$) classifier $f(\boldsymbol{x}) = g(h(\boldsymbol{x}))$, which uses a differentiable scoring function $h : \mathcal{X} \to [0, 1]$. Examples of such scoring functions include those used in logistic regression, random forests with class probabilities, or deep models with a softmax top layer. It is worth noting that such scoring functions are often interpreted as estimating the probability of a positive outcome given the input data. $g : [0, 1] \to \mathcal{Y}$ is an activation function that maps scores to binary labels. Throughout the remainder of the paper, we will use $g(t) = \boldsymbol{I}_{t \geq \eta}$, in which $\eta$ is the decision threshold with respect to the given scoring function $h$ and we assume $\eta > 1/2$. In addition to its use as a probability scoring function, the function $h(\cdot)$ can be utilized to define the nonconformity score function (Johansson et al., 2017). We present the definition of the nonconformity function below, accompanied by a remark to clarify its underlying intuition.

**Definition 1.** The nonconformity score function $s(\boldsymbol{x}, y)$ is defined as

$$s(\boldsymbol{x}, y) = (1 - y)h(\boldsymbol{x}) + y(1 - h(\boldsymbol{x})).$$

*Remark.* The nonconformity score, denoted by an arbitrary function $s : \mathcal{X} \times \mathcal{Y} \to \mathbb{R}$, quantifies the degree of strangeness of an example $(\boldsymbol{x}, y)$ (Johansson et al., 2017). In predictive modeling, nonconformity functions are often constructed using the prediction error of a classification model. For instance, $s(\boldsymbol{x}, y) = \Delta[f(\boldsymbol{x}, y)]$, where $\Delta$ is an error metric. This is based on the idea that uncommon or unusual examples tend to have larger prediction errors than common or normal examples. For classification problems, one can define the nonconformity function as an error function $\Delta$ applied to the probability estimates $h$ provided by the model $f$, i.e. $\Delta[f(\boldsymbol{x}, y)] = 1 - h(y|\boldsymbol{x})$ (Johansson et al., 2017). The resulting definitions are provided in Definition 1.

As shown in Figure 1, when data shifts over time, the predictive model changes correspondingly from $f(\cdot) = g(h(\cdot))$ to $f'(\cdot) = g(h'(\cdot))$ (at $t = t_1$), where the activation function $g(\cdot)$ remains the same, $h'$ and $f'$ are unknown. To constrain the model variations, we make the following assumption.

**Assumption 2.** We assume that the scoring function of the model before and after change exhibits a small perturbation captured by a small number $\tau$, such that $h(\boldsymbol{x}) - \tau \leq h'(\boldsymbol{x}) \leq h(\boldsymbol{x}) + \tau, \forall \boldsymbol{x} \in \mathcal{X}$. The value of $\tau$ can be set based on the magnitude of the expected shifts and the sensitivity of the deployed model to distributional shifts.

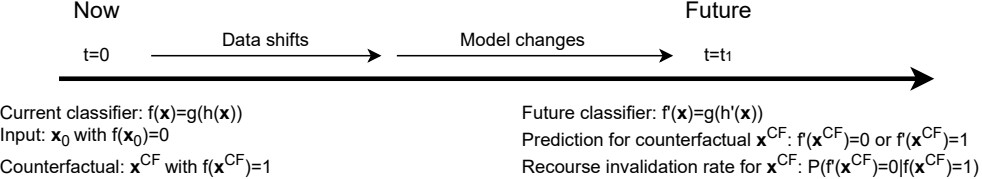

Figure 1: Setup of recourse action under distributional and model shifts

*Remark.* Our decision to introduce the above assumption is based on common occurrences in dynamic decision systems, such as model updates or changes in data distribution. We recognize that these shifts are typically not extreme, leading to our theoretical assumption that perturbations are bounded by $\tau$. Specifically, it constrains the future classifier's score function $h'(\boldsymbol{x})$ to a small perturbation range around the current score function $h(\boldsymbol{x})$. We can define the constraint set as $\mathcal{C}_f = \{f' : h(\boldsymbol{x}) - \tau \leq h'(\boldsymbol{x}) \leq h(\boldsymbol{x}) + \tau, \forall \boldsymbol{x} \in \mathcal{X}\}$. Our theoretical analysis is focused on classifiers $f'$ within this constraint set, i.e. $f' \in \mathcal{C}_f$.

We now compare our assumption with the assumptions made in (Upadhyay et al., 2021) and (Nguyen et al., 2023). In (Upadhyay et al., 2021), $\mathcal{M}$ represents the current predictive model, and $\mathcal{M}_\delta$ denotes the shifted model resulting from applying an operation $\delta$ to either the parameter space or the gradient space. When $\delta$ operates on the parameter space, it corresponds to small model shifts within a limited range of the parameter space. When $\delta$ operates on the gradient space, it restricts model shifts within a norm-ball. Nguyen et al. (2023) also consider parametric predictive models and proposes a framework for handling stochastic changes in the parameters of the considered models. Specifically, the shifted parameters are represented by a random vector $\tilde{\boldsymbol{\theta}}$ that follows from a finite mixture of distributions with $K$ components, i.e., $\tilde{\boldsymbol{\theta}} \sim (\hat{\boldsymbol{\theta}}_k, \hat{\boldsymbol{\sigma}}_k, \hat{p}_k)_{k \in [K]}$. Each component in the mixture represents one specific type of model shift, and the weight $\hat{p}_k$ reflects the proportion of the $k$-th shift type. Although Gaussian mixture models have been shown to be effective approximators of densities, the proposed mixture model in (Nguyen et al., 2023) only performs well against a large number of distributional perturbations when the number of components is large enough. Furthermore, selecting appropriate components for different application scenarios can also be a challenging task. Unlike the assumptions in these two papers, our constraint on model shifts relies solely on the value of the scoring function, and applies to both parametric and non-parametric models.

As the predictive model changes, we can express the updated nonconformity function associated with the new model $f'$ as $s'(\boldsymbol{x}, y)$, where $s(\boldsymbol{x}, y) = (1 - y)h(\boldsymbol{x}) + y(1 - h(\boldsymbol{x}))$.

We now introduce the concept of recourse invalidation rate to quantify the probability that a recourse becomes invalid under model shifts. Let $\boldsymbol{x}_0$ be an instance with an unfavorable prediction by the current model, i.e. $f(\boldsymbol{x}_0) = 0$ and $h(\boldsymbol{x}_0) < \eta$. Let $\boldsymbol{x}^{\mathrm{CF}}$ be a recourse for $\boldsymbol{x}_0$ generated by any state-of-the-art algorithmic recourse framework such that $f(\boldsymbol{x}^{\mathrm{CF}}) = 1$ and $h(\boldsymbol{x}^{\mathrm{CF}}) \geq \eta$. The robustness of $\boldsymbol{x}^{\mathrm{CF}}$ under model shift is measured by the recourse invalidation rate.

**Definition 3.** The recourse invalidation rate under model shift is defined as $r_{\mathrm{ivd}}(\boldsymbol{x}^{\mathrm{CF}}) = \mathbb{P}(f'(\boldsymbol{x}^{\mathrm{CF}}) = 0 | f(\boldsymbol{x}^{\mathrm{CF}}) = 1)$.

*Remark.* The probability $\mathbb{P}(f'(\boldsymbol{x}^{\mathrm{CF}}) = 0 | f(\boldsymbol{x}^{\mathrm{CF}}) = 1)$ is taken over the distribution of a future classifier $f'$. As shown in Figure 1, following the deployment of the current classifier, future data shifts may occur. In response to these shifts, the model would be retrained using newly collected data, leading to a new classifier $f'$. In practical scenarios, the specifics of future data shifts and the consequent updated model $f'$ are typically unknown. Thus, $f'$ is treated as a random function in our analysis. The randomness in $f'$ comes from possible future data shifts and the associated model adaptations. To address the inherent uncertainty, Assumption 2 constrains the variation in the score function post-shift.

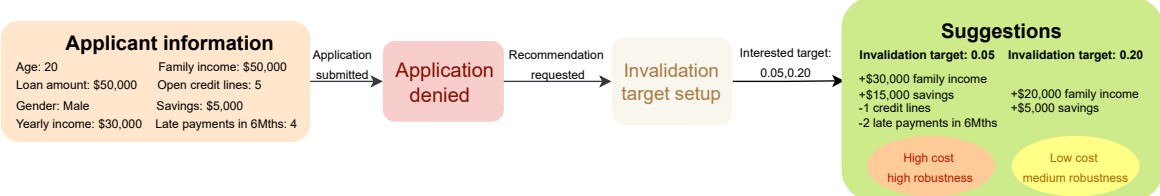

Figure 2: Loan application example: a practical view on the cost-robustness trade-off.

Using this metric, this paper aims to answer two questions: 1) *Recourse invalidation rate estimation*: For a given recourse $\boldsymbol{x}^{\mathrm{CF}}$, can we provide a reasonable estimate of $r_{\mathrm{ivd}}(\boldsymbol{x}^{\mathrm{CF}})$ under a shifted model $f'$?; 2) *Recourse generation with given invalidation rate constraint:* Suppose a user provides a tolerance on recourse invalidation rate, how can we generate a minimal cost recourse that satisfies such constraint?

To answer the first question, we provide an upper-bound for the recourse invalidation rate $r_{\mathrm{ivd}}(\boldsymbol{x}^{\mathrm{CF}})$. More precisely, we aim to derive $r_u(\cdot)$ so that $r_{\mathrm{ivd}}(\boldsymbol{x}^{\mathrm{CF}}) \leq r_u(\boldsymbol{x}^{\mathrm{CF}})$.

For the recourse generation with constrained invalidation rate problem, we define a cost function $c : \mathbb{R}^d \to \mathbb{R}_+$ that measures the cost of a given recourse (Rawal et al., 2021; Upadhyay et al., 2021; Mothilal et al., 2020). To satisfy the user's preference on the invalidation rate, we aim to solve the optimization problem

$$\min_{\boldsymbol{x}^{\mathrm{CF}} \in \mathcal{A}} c(\boldsymbol{x}_0, \boldsymbol{x}^{\mathrm{CF}}), \ \text{s.t.} \ f(\boldsymbol{x}^{\mathrm{CF}}) = 1, r_u(\boldsymbol{x}^{\mathrm{CF}}) \leq \gamma_t, \tag{1}$$

where $\gamma_t$ is the maximum allowed invalidation rate determined by the user. In particular, the objective function encourages the implementation cost to be lower, the first constraint ensures that the recourse is valid under the current model and the second constraint upper-bounds the recourse invalidation probability as model shifts. For example, in the context of loan applications, as shown in Figure 2, the user may request recommendations from the bank upon being denied. Once invalidation targets are set, recommendations with varying levels of cost and robustness will be provided.

## 4 Recourse Invalidation Estimation for a Given Recourse

In this section, for a pre-computed recourse $\boldsymbol{x}^{\mathrm{CF}}$, we propose an uncertainty quantification method to calculate theoretical upper-bounds of the invalidation rate $r_{\mathrm{ivd}}(\boldsymbol{x}^{\mathrm{CF}})$ under potential model shifts.

We first provide the formula of the upper-bound for $r_{\mathrm{ivd}}(\boldsymbol{x}^{\mathrm{CF}})$.

**Lemma 4.** *For the recourse invalidation rate, we have*

$$r_{ivd}(\boldsymbol{x}^{CF}) \ = \ \mathbb{P}(f'(\boldsymbol{x}^{CF}) = 0 | f(\boldsymbol{x}^{CF}) = 1) \leq 1 - \frac{\mathbb{P}(s'(\boldsymbol{x}^{CF}, 1) \leq 1 - \eta)}{\mathbb{P}(s'(\boldsymbol{x}^{CF}, 1) \leq 1 - \eta + \tau)}.$$

To compute this upper-bound, we need to analyze the value of $s'(\boldsymbol{x}^{\mathrm{CF}}, 1)$, which quantifies the degree to which the counterfactual sample $(\boldsymbol{x}^{\mathrm{CF}}, 1)$ conforms to calibration samples in $\mathcal{D}_{\mathrm{calib}}$. However, the nonconformity scores $\{s(\boldsymbol{X}_i, Y_i)\}_{i=1}^n$ for samples in $\mathcal{D}_{\mathrm{calib}}$ are calculated based on the current model. As such, we must establish a connection between $s$ and $s'$ to proceed with our analysis.

We conceptualize the nonconformity function $s(\cdot, \cdot)$ as a nonconformity random variable $S \in [0, 1]$. Under the current model, for samples $(\boldsymbol{X}, Y)$ in $\mathcal{D}$, we assume that $S = s(\boldsymbol{X}, Y) \sim P$ with $p(\cdot)$ as the probability density function (pdf). As the model shifts to $f'$, we assume that $S = s'(\boldsymbol{X}, Y) \sim P'$, with $p'(\cdot)$ as the pdf. To constrain the perturbation of $P'$ from $P$ within a certain bounded distance, we introduce the following definitions.

**Definition 5.** Define point-wise likelihood ratio between $P$ and $P'$ as $v(s) = \frac{p'(s)}{p(s)}$. Define the identification set as $\mathcal{P}(P, L, U) = \{P' : L(s) \leq v(s) \leq U(s) \ P\text{-almost surely}\}$, where $L(\cdot)$ and $U(\cdot)$ are pre-specified functions serving as point-wise lower and upper-bounds for likelihood ratio.

In light of Assumption 2 that limits the deviation of the scoring function value between the current and potential future models, we convert the conditions in Assumption 2 to conditions on $P$ and $P'$ and specify the identification set $\mathcal{P}(P, L, U)$ based on samples in the training set $\mathcal{D}_{\text{train}}$.

**Lemma 6.** *Under Assumption 2, we have $\int_{s-\delta}^{s+\delta} p'(t)dt \leq \int_{s-\tau-\delta}^{s+\tau+\delta} p(t)dt$.*

Using this lemma, we construct $\hat{L}$ and $\hat{U}$ with $\hat{L}(s) \leq \hat{U}(s), \forall s \in [0,1]$ and use them as the pointwise lower and upper-bounds for the unknown likelihood ratio $v(s)$. It is worth noting that the proposed theoretical guarantee does not require perfect bounds and is valid even when there are expectations (as we will see later in Propositions 7 and 8). We can construct $\hat{L}$ and $\hat{U}$ as follows:
1. derive the empirical probability mass function $\hat{p}$ of $P$ and denote the support of $\hat{p}$ as $\mathcal{S}_{\text{train}}$;
2. choose a parameter $\epsilon$ such that $\epsilon \leq \sum\limits_{\{s' \in \mathcal{S}_{\text{train}}: s-\tau \leq s' \leq s+\tau\}} \hat{p}(s'), \forall s \in \mathcal{S}_{\text{train}}$ (which guarantees $\hat{L}(s) \leq \hat{U}(s)$),

e.g. we can set $\epsilon = \min\limits_{\{s \in \mathcal{S}_{\text{train}}\}} \hat{p}(s)$;

3. let $\hat{L}(s) = \frac{\epsilon}{\hat{p}(s)}$, $\hat{U}(s) = \frac{\sum\limits_{\{s' \in \mathcal{S}_{\text{train}}: s-\tau \leq s' \leq s+\tau\}} \hat{p}(s')}{\hat{p}(s)}$.

Based on the constructed bounds $\hat{L}(\cdot)$ and $\hat{U}(\cdot)$, we use the calibration fold to quantify the nonconformity of $s'(\boldsymbol{x}^{\text{CF}}, 1)$. For samples in $\mathcal{D}_{\text{calib}}$, let $S_i = s(\boldsymbol{X}_i, Y_i)$, and denote $[1], \cdots, [n]$ as a permutation of $\{1, \cdots, n\}$ such that $S_{[1]} \leq \cdots \leq S_{[n]}$. Let $L_i = \hat{L}(S_i), U_i = \hat{U}(S_i)$. For $\boldsymbol{x}^{\text{CF}}$, denote $L^{\text{CF}} = \min\{\hat{L}(s(\boldsymbol{x}^{\text{CF}}, 0)), \hat{L}(s(\boldsymbol{x}^{\text{CF}}, 1))\}, U^{\text{CF}} = \max\{\hat{U}(s(\boldsymbol{x}^{\text{CF}}, 0)), \hat{U}(s(\boldsymbol{x}^{\text{CF}}, 1))\}$. Define

$$\hat{F}(k) = \frac{\sum\limits_{i=1}^{k} L_{[i]}}{\sum\limits_{i=1}^{k} L_{[i]} + \sum\limits_{i=k+1}^{n} U_{[i]} + U^{\text{CF}}}, \quad \hat{E}(t) = \frac{\sum\limits_{i=1}^{t} U_{[i]}}{\sum\limits_{i=1}^{t} U_{[i]} + \sum\limits_{i=t+1}^{n} L_{[i]} + L^{\text{CF}}}.$$

Then we have the following two propositions characterizing the coverage guarantee of $s'(\boldsymbol{x}^{\text{CF}}, y^{\text{CF}})$, where $y^{\text{CF}} = f'(x^{\text{CF}})$, $\|Z\|_r = (\mathbb{E}[|Z|^r])^{1/r}$ is the $L_r$ norm for any random variable $Z$ with $r \geq 1$.

**Proposition 7.** *For a given $\alpha$, we have $\mathbb{P}(s'(\boldsymbol{x}^{CF}, y^{CF}) \leq S_{[k^*]}) \geq 1 - \alpha - \hat{\Delta}_F$, where $k^* = \min\{k : \hat{F}(k) \geq 1 - \alpha\}$, $\hat{\Delta}_F = \left\|\frac{1}{\hat{L}(S)}\right\|_q \cdot \left\|\max\{0, \hat{L}(S) - v(S)\}\right\|_p$ with $\frac{1}{p} + \frac{1}{q} = 1$.*

*Remark.* $\hat{F}$ is an increasing function of $k$, which indicates that a lot of search methods can be applied to find $k^*$ given a target level $\alpha$ (an algorithm is provided in the Appendix B.1). If $\hat{L}(s) \leq v(s)$ *a.s.*, then $\hat{\Delta}_F = 0$ and $\mathbb{P}(s'(\boldsymbol{x}^{\text{CF}}, y^{\text{CF}}) \leq S_{[k^*]}) \geq 1 - \alpha$.

**Proposition 8.** *For a given $\beta$, we have $\mathbb{P}(s'(\boldsymbol{x}^{CF}, y^{CF}) \leq S_{[t^*]}) \leq 1 - \beta + \hat{\Delta}_E$, where $t^* = \max\{t : \hat{E}(t) \leq 1 - \beta\}$, $\hat{\Delta}_E = \left\|\frac{1}{\hat{U}(S)}\right\|_q \cdot (\left\|\max\{0, \hat{L}(S) - v(S)\}\right\|_p + \frac{1}{n}\|v(S)^{1/p} \max\{\hat{L}(S) - v(S), 0\}\|_p)$.*

*Remark.* For $\frac{a_1}{a_2} < 1$, we have $\forall 0 < \delta < \infty, \frac{a_1+\delta}{a_2+\delta} > \frac{a_1}{a_2}$, which indicates that $\frac{a_1+\delta_1}{a_2+\delta_2} > \frac{a_1+\delta}{a_2+\delta} > \frac{a_1}{a_2}$, if $\delta_1 > \delta_2 > 0$. Thus, $\hat{E}$ is an increasing function of $t$, and search methods can be applied to find $t^*$ easily. If $\hat{L}(s) \leq v(s)$ *a.s.*, then $\hat{\Delta}_E = 0$ and $\mathbb{P}(s'(\boldsymbol{x}^{\text{CF}}, y^{\text{CF}}) \leq S_{[t^*]}) \leq 1 - \beta$.

We then provide two upper-bounds for $r_{\text{ivd}}(\boldsymbol{x}^{\text{CF}})$. By using Propositions 7 and 8, we can derive the upper-bound in Lemma 4. We begin by presenting a weaker bound that is easier to compute.

**Theorem 9.** *Based on the assumed shift parameter $\tau$, we have the following two cases.*
*1. If $\tau$ is small s.t. $\tau < 2\eta - 1$, then we have $r_{ivd}(\boldsymbol{x}^{CF}) \leq \alpha_1 + \hat{\Delta}_F$, where $\alpha_1 = 1 - \hat{F}(k_1^*)$ with $k_1^* = \max\{k_1 : S_{[k_1]} \leq 1 - \eta\}$. The procedure for finding $\alpha_1$ and $k_1^*$ is summarized in Algorithm 1. 2. If $\tau \geq 2\eta - 1$, then as long as $h(\boldsymbol{x}^{CF}) > 1 - \eta + \tau$, we still have $r_{ivd}(\boldsymbol{x}^{CF}) \leq \alpha_1 + \hat{\Delta}_F$.*

Next, we present an alternative bound that is tighter than the previous one.

**Theorem 10.** *The upper-bound in Theorem 9 can be refined as $r_{ivd}(\boldsymbol{x}^{CF}) \leq 1 - \frac{1-\alpha_1-\hat{\Delta}_F}{1-\alpha_2+\hat{\Delta}_E}$, where $\alpha_2 = 1 - \hat{E}(k_2^*)$ with $k_2^* = \min\{k_2 : S_{[k_2]} \geq 1 - \eta + \tau\}$.*

Compared to the lower-bound analysis presented in (Upadhyay et al., 2021), which aims to demonstrate the likelihood of invalidation of recourses generated without accounting for model shifts, our approach involves an upper-bound on $r_{\mathrm{ivd}}$, thereby providing a threshold for the acceptable level of invalidation and facilitating the design of recourses with limited invalidation rates. Additionally, the lower bound analysis in (Upadhyay et al., 2021) assumes a Gaussian distribution for $\boldsymbol{x}^{\mathrm{CF}}$, which is unlikely to hold in real-world scenarios, as well as a strong assumption regarding the model shift parameter with respect to the proposed Gaussian distribution. On the contrary, in Theorems 9 and 10, we do not make any distributional assumptions on $\boldsymbol{X}/\boldsymbol{x}^{\mathrm{CF}}$ and the constraint on the model shift parameter is less restrictive.

Bui et al. (2022) also propose an uncertainty quantification tool to compute the lower and upper-bounds of the validation probability. However, the bounds derived in (Bui et al., 2022) are restricted to linear classifiers and require knowledge of the first- and second-moment information of the nominal distribution assumed for the model parameter. Furthermore, they rely on the use of Gelbrich distances to define the set of model uncertainties in order to efficiently obtain the bounds by solving semidefinite programs. In contrast, our proposed invalidation upper-bounds can be applied to any model, including non-parametric models, and can be efficiently computed using Algorithm 1.

## 5 Recourse Generation with Invalidation Rate Constraint

Arming with the ability to upper-bound the recourse invalidation rates, in this section, we propose an algorithm for users to manage the trade-off between recourse cost and the risk of the recourse being invalidated under model shifts.

Suppose the user-specified invalidation rates are $\gamma_1, \gamma_2, \cdots, \gamma_m$. For one specific invalidation rate $\gamma_t$, the goal is to find a recourse $\boldsymbol{x}^{\mathrm{CF}}$ whose probability of becoming invalid under potential model shifts is at most $\gamma_t$. To bound the recourse invalidation rate, we require $\gamma_t \geq r_u(\boldsymbol{x}^{\mathrm{CF}})$, and that the recourse should be valid under the current model, i.e. $f(\boldsymbol{x}^{\mathrm{CF}}) = 1$.

To design an algorithm that guides the recourse search towards regions of low cost while satisfying the invalidation constraint, we solve the optimization problem in (1). Here, the constraint about recourse invalidation rate relies on the value of $\alpha_1$ (and $\alpha_2$ for the tighter bound), which is simulation-based. To address this, we apply iterative optimization approaches that do not rely on derivatives. In the following, we provide an outline of the proposed method. We name our algorithm probabilistic invalidation based robust recourse (PiRR) generation algorithm.

For our proposed optimization problem (1), it is a nonconvex problem with both equality and inequality constraints. We will use extended ADMM algorithms that can be applied to solve constrained nonconvex problems efficiently (Wang et al., 2019; Themelis & Patrinos, 2020). First use the penalty function and variable substitution to replace the inequality constraint. Then (1) can be rewritten as

$$\min_{\boldsymbol{x}^{\mathrm{CF}} \in \mathcal{A}, z} c(\boldsymbol{x}_0, \boldsymbol{x}^{\mathrm{CF}}) + I(z), \ \text{s.t.} f'(\boldsymbol{x}^{\mathrm{CF}}) = f(\boldsymbol{x}^{\mathrm{CF}}) - 1 = 0, \ r'(\boldsymbol{x}^{\mathrm{CF}}, z) = r_u(\boldsymbol{x}^{\mathrm{CF}}) - z = 0, \tag{2}$$

---

**Algorithm 1** Procedure of finding $\alpha_1, k_1^*$ in $r_u(\boldsymbol{x}^{\mathrm{CF}})$

---

**Input:** $\mathcal{D}_{\mathrm{train}}, \mathcal{D}_{\mathrm{calib}}$, recourse $\boldsymbol{x}^{\mathrm{CF}}$, nonconformity score function $s(\cdot, \cdot)$, model shift parameter $\tau$.
 1: Use $\mathcal{D}_{\mathrm{train}}$ to construct the point-wise bounds $\hat{L}(\cdot)$ and $\hat{U}(\cdot)$ for likelihood ratio $v$.
 2: For samples in $\mathcal{D}_{\mathrm{calib}}$, compute $S_i = s(\boldsymbol{X}_i, Y_i), L_i = \hat{L}(S_i), U_i = \hat{U}(S_i)$.
 3: For recourse $\boldsymbol{x}^{\mathrm{CF}}$, compute $L^{\mathrm{CF}}$ and $U^{\mathrm{CF}}$.
 4: For $1 \leq k \leq n$, compute $\hat{F}(k)$.
 5: Derive the value of $k_1^*$ from $k_1^* = \max\{k_1 : S_{[k_1]} \leq 1 - \eta\}$.
 6: Compute $\alpha_1 = 1 - \hat{F}(k_1^*)$.
**Output:** The value of $\alpha_1, k_1^*$.

---

where $l(z) = \infty \cdot \mathbf{1}_{z > \gamma_t}$. The equality constrained problem (2) can be further transformed to its Augmented Lagrangian problem:

$$\mathcal{L}_\rho(\boldsymbol{x}^{\mathrm{CF}}, z, \mu_1, \mu_2) = c(\boldsymbol{x}_0, \boldsymbol{x}^{\mathrm{CF}}) + I(z) + \mu_1 f'(\boldsymbol{x}^{\mathrm{CF}}) + \mu_2 r'(\boldsymbol{x}^{\mathrm{CF}}, z) + \frac{\rho}{2} \left[ f'(\boldsymbol{x}^{\mathrm{CF}})^2 + r'(\boldsymbol{x}^{\mathrm{CF}}, z)^2 \right],$$

where $\mu_1, \mu_2$ are Lagrangian multipliers, and $\rho > 0$ is a penalty constant. Using ADMM, there are 3 update steps:

1. $\boldsymbol{x}^{\mathrm{CF}}$ update: $(\boldsymbol{x}^{\mathrm{CF}})^{k+1} = \arg \min_{\boldsymbol{x}^{\mathrm{CF}} \in \mathcal{A}} \mathcal{L}_\rho(\boldsymbol{x}^{\mathrm{CF}}, z^k, \mu_1^k, \mu_2^k);$

2. $z$ update: $z^{k+1} = \max\{0, \min\{\gamma_t, \arg \min_z \mathcal{L}_\rho((\boldsymbol{x}^{\mathrm{CF}})^{k+1}, z, \mu_1^k, \mu_2^k)\}\};$

3. multipliers update: $\mu_1^{k+1} = \mu_1^k + \rho f'((\boldsymbol{x}^{\mathrm{CF}})^{k+1}), \mu_2^{k+1} = \mu_2^k + \rho r'((\boldsymbol{x}^{\mathrm{CF}})^{k+1}, z^{k+1}).$

In particular, for the sub-problem on $\boldsymbol{x}^{\mathrm{CF}}$, since $r_u(\boldsymbol{x}^{\mathrm{CF}})$ can only be evaluated based on simulations, its gradient information can only be obtained by numerical differentiation. Thus, we apply derivative-free optimization methods (Larson et al., 2019; Shi et al., 2021) as this kind of approaches only require the objective function value and constraint function values to be evaluated. For the sub-problem on $z$, we note that on the interval $[0, \gamma_t]$, it is a quadratic optimization problem on $z$ and can be easily solved. Since ADMM takes the form of a decomposition-coordination procedure, the solutions to small local subproblems are coordinated to find a solution to the large global problem. The proposed ADMM based PiRR procedure is summarized in Algorithm 2.

## 6 Numerical Results

This section presents numerical examples to provide empirical validation of our proposed theoretical framework. In Section 6.1, we demonstrate the efficacy of our approach for estimating the recourse invalidation rate based on state-of-the-art recourse generating algorithms. In Section 6.2, we evaluate the effectiveness of our proposed algorithms in finding recourses that satisfy prescribed invalidation targets. Additional numerical results are available in Appendix C.

**Datasets:** To evaluate the effect of distributional shifts on the validity of generated recourses, we analyze three datasets from distinct domains, namely the Criminal justice dataset (Lakkaraju et al., 2016), the Student performance dataset (Amrieh et al., 2016) and the German credit dataset (Dua & Graff, 2017). Each dataset consists of two parts, initial data $D_1$ and shifted data $D_2$, and exhibits various types of distributional shifts. Details can be found in Appendix C.

**Predictive models:** We consider two predictive models: neural network (NN) and logistic regression (LR). In particular, for NN models, we train the ReLu-based models with 50 hidden layers.

**Cost functions:** We use two cost functions to measure the recourse implementation cost, $\ell_1$ distance and a cost function learned from pairwise feature comparison inputs (PFC) (Rawal et al., 2021).

**Baseline recourse algorithms:** We consider three baseline recourse generating methods to validate the theoretical bounds on recourse invalidation rate: counterfactual explanations (CF) framework outlined by (Wachter et al., 2017), actionable recourse (AR) in linear classification (Ustun et al., 2019), and causal

---

**Algorithm 2** PiRR: Probabilistic Invalidation based Robust Recourse Generation

**Input:** $\boldsymbol{x}_0$, current model $f$, model shift parameter $\tau$, invalidation rate $\gamma_t$, penalty constant $\rho$.
1: initialize: $(\boldsymbol{x}^{\mathrm{CF}})^0 = \boldsymbol{x}_0, z^0 = \gamma_t, \mu_1^0 = 0, \mu_2^0 = 0, k = 0;$
2: **repeat**
3:    $(\boldsymbol{x}^{\mathrm{CF}})^{k+1} = \arg \min_{\boldsymbol{x}^{\mathrm{CF}} \in \mathcal{A}} \mathcal{L}_\rho(\boldsymbol{x}^{\mathrm{CF}}, z^k, \mu_1^k, \mu_2^k);$
4:    $z^{k+1} = \max\{0, \min\{\gamma_t, \arg \min_z \mathcal{L}_\rho(\boldsymbol{x}^{k+1}, z, \mu_1^k, \mu_2^k)\}\};$
5:    $\mu_1^{k+1} = \mu_1^k + \rho f'((\boldsymbol{x}^{\mathrm{CF}})^{k+1});$
6:    $\mu_2^{k+1} = \mu_2^k + \rho r'((\boldsymbol{x}^{\mathrm{CF}})^{k+1}, z^{k+1});$
7:    $k = k + 1;$
8: **until** convergence
**Output:** $(\boldsymbol{x}^{\mathrm{CF}})^* = (\boldsymbol{x}^{\mathrm{CF}})^k.$

---

Table 1: Theoretical and empirical recourse invalidation ($\ell_1$ cost, $\epsilon = \min_{\{s \in \mathcal{S}_{\text{train}}\}} \hat{p}(s)$).

| Algorithm | Dataset | Predictive model | Upper-bound in Theorem 9 | Upper-bound in Theorem 10 | Empirical invalidation rate |
|---|---|---|---|---|---|
| CF | Criminal justice | LR | $0.81 \pm 0.04$ | $0.75 \pm 0.05$ | 0.70 |
| | | NN | $0.66 \pm 0.05$ | $0.59 \pm 0.10$ | 0.49 |
| | Student performance | LR | $0.82 \pm 0.07$ | $0.79 \pm 0.09$ | 0.72 |
| | | NN | $0.70 \pm 0.08$ | $0.62 \pm 0.11$ | 0.53 |
| | German credit | LR | $0.63 \pm 0.04$ | $0.57 \pm 0.09$ | 0.44 |
| | | NN | $0.63 \pm 0.05$ | $0.60 \pm 0.07$ | 0.54 |
| AR | Criminal justice | LR | $0.91 \pm 0.03$ | $0.89 \pm 0.04$ | 0.85 |
| | | NN | $0.54 \pm 0.10$ | $0.45 \pm 0.14$ | 0.34 |
| | Student performance | LR | $0.70 \pm 0.08$ | $0.67 \pm 0.10$ | 0.59 |
| | | NN | $0.28 \pm 0.03$ | $0.19 \pm 0.04$ | 0.16 |
| | German credit | LR | $0.56 \pm 0.08$ | $0.54 \pm 0.11$ | 0.46 |
| | | NN | $0.78 \pm 0.05$ | $0.74 \pm 0.05$ | 0.68 |
| MINT | German credit | LR | $0.18 \pm 0.04$ | $0.13 \pm 0.05$ | 0.09 |
| | | NN | $0.53 \pm 0.04$ | $0.44 \pm 0.07$ | 0.38 |

Table 2: Theoretical and empirical recourse invalidation (PFC cost, $\epsilon = \min_{\{s \in \mathcal{S}_{\text{train}}\}} \hat{p}(s)$)

| Algorithm | Dataset | Predictive model | Upper-bound in Theorem 9 | Upper-bound in Theorem 10 | Empirical invalidation rate |
|---|---|---|---|---|---|
| CF | Criminal justice | LR | $0.85 \pm 0.04$ | $0.79 \pm 0.05$ | 0.73 |
| | | NN | $0.67 \pm 0.06$ | $0.58 \pm 0.10$ | 0.51 |
| | Student performance | LR | $0.91 \pm 0.08$ | $0.88 \pm 0.09$ | 0.82 |
| | | NN | $0.83 \pm 0.08$ | $0.78 \pm 0.11$ | 0.69 |
| | German credit | LR | $0.64 \pm 0.06$ | $0.59 \pm 0.06$ | 0.43 |
| | | NN | $0.66 \pm 0.06$ | $0.64 \pm 0.07$ | 0.50 |
| AR | Criminal justice | LR | $0.93 \pm 0.04$ | $0.92 \pm 0.04$ | 0.90 |
| | | NN | $0.77 \pm 0.03$ | $0.69 \pm 0.04$ | 0.65 |
| | Student performance | LR | $0.85 \pm 0.08$ | $0.81 \pm 0.09$ | 0.76 |
| | | NN | $0.28 \pm 0.04$ | $0.22 \pm 0.04$ | 0.19 |
| | German credit | LR | $0.64 \pm 0.08$ | $0.56 \pm 0.11$ | 0.50 |
| | | NN | $0.59 \pm 0.04$ | $0.54 \pm 0.05$ | 0.42 |
| MINT | German credit | LR | $0.25 \pm 0.07$ | $0.16 \pm 0.08$ | 0.06 |
| | | NN | $0.51 \pm 0.08$ | $0.45 \pm 0.09$ | 0.37 |

recourse framework (MINT) proposed by (Karimi et al., 2020a). In addition, we compare our proposed robust recourse generating methods with several other methods that utilize robustness. These include ROAR, which generates recourses that are robust to model updates (Upadhyay et al., 2021); ARAR, which finds recourses robust to uncertainty in the inputs (Dominguez-Olmedo et al., 2022); PROBE, which allows users to choose the invalidation rate if small changes are made to the recourse (Pawelczyk et al., 2022); DiRRAc, which handles stochastic changes in the model parameters (Nguyen et al., 2023).

**Experimental setting:** We use 5-fold cross validation throughout the experiments. On $D_1$, we use 4 folds (training fold) to train predictive models and the remaining fold (validation/calibration fold) to generate recourses. On $D_2$, we train the shifted models. The retrained model is applied on the validation fold to obtain the empirical recourse invalidation rate.

## 6.1 Validating the recourse invalidation bounds

Since the generated recourse for each negatively-predicted sample is different, the derived bounds on the recourse invalidation rate are also different. To evaluate the theoretical bounds, we compute the average upper-bounds as well as the standard deviation. For $\ell_1$ cost, the results are provided in Table 1. For the

Table 3: PiRR compared to existing robust recourse generating methods (LR model, $\ell_1$ cost).

| Dataset | Algorithm | Invalidation rate before shift | Invalidation rate after shift | Average cost |
|---|---|---|---|---|
| | ROAR | $0.00 \pm 0.00$ | $0.02 \pm 0.02$ | $3.14 \pm 0.25$ |
| | ARAR | $0.00 \pm 0.00$ | $0.03 \pm 0.02$ | $2.07 \pm 0.31$ |
| | PROBE | $0.00 \pm 0.00$ | $0.02 \pm 0.02$ | $1.76 \pm 0.35$ |
| Criminal justice | DiRRAc | $0.00 \pm 0.00$ | $\mathbf{0.02 \pm 0.02}$ | $1.68 \pm 0.32$ |
| | PiRR(0.10) | $\mathbf{0.00 \pm 0.00}$ | $0.07 \pm 0.02$ | $\mathbf{1.03 \pm 0.25}$ |
| | PiRR(0.05) | $\mathbf{0.00 \pm 0.00}$ | $0.03 \pm 0.01$ | $\mathbf{1.27 \pm 0.32}$ |
| | ROAR | $0.00 \pm 0.00$ | $0.06 \pm 0.10$ | $2.02 \pm 0.38$ |
| | ARAR | $0.00 \pm 0.00$ | $0.05 \pm 0.10$ | $1.54 \pm 0.40$ |
| | PROBE | $0.00 \pm 0.00$ | $0.09 \pm 0.08$ | $1.37 \pm 0.35$ |
| Student performance | DiRRAc | $0.00 \pm 0.00$ | $0.05 \pm 0.09$ | $1.55 \pm 0.34$ |
| | PiRR(0.10) | $\mathbf{0.00 \pm 0.00}$ | $0.08 \pm 0.02$ | $\mathbf{1.29 \pm 0.35}$ |
| | PiRR(0.05) | $\mathbf{0.00 \pm 0.00}$ | $\mathbf{0.04 \pm 0.01}$ | $\mathbf{1.43 \pm 0.42}$ |
| | ROAR | $0.00 \pm 0.00$ | $0.06 \pm 0.15$ | $3.88 \pm 0.54$ |
| | ARAR | $0.00 \pm 0.00$ | $0.04 \pm 0.08$ | $2.27 \pm 0.36$ |
| | PROBE | $0.00 \pm 0.00$ | $0.03 \pm 0.07$ | $\mathbf{1.54 \pm 0.37}$ |
| German credit | DiRRAc | $0.00 \pm 0.00$ | $\mathbf{0.01 \pm 0.06}$ | $1.62 \pm 0.30$ |
| | PiRR(0.10) | $\mathbf{0.00 \pm 0.00}$ | $0.07 \pm 0.02$ | $\mathbf{1.34 \pm 0.36}$ |
| | PiRR(0.05) | $\mathbf{0.00 \pm 0.00}$ | $\mathbf{0.03 \pm 0.02}$ | $1.57 \pm 0.39$ |

PFC cost, the results are provided in Table 2. Our analysis shows that as the predictive model shifts, the generated recourses have a high probability of becoming invalid. The invalidation rate depends on various factors such as the choice of predictive model, the shifts in the dataset, and the recourse generating algorithms used. Among the three baseline methods, MINT is shown to have better robustness when the model shifts because it leverages the causal graph to generate recourse. Regarding the derived theoretical bounds on the recourse invalidation rate, our results demonstrate that the empirical invalidation rates always lie below the theoretical upper-bounds, and that the upper-bound obtained from Theorem 10 is tighter with a slightly larger deviation than that obtained from Theorem 9. Furthermore, we observe that for a given recourse generating method, the theoretical bounds for recourses of different samples do not deviate significantly. The reason for this is that all considered baseline algorithms aim to find low-cost recourses, which are likely to lie on the decision boundary and have similar robustness to model changes.

### 6.2 Effectiveness of proposed algorithms

We evaluate the performance of PiRR against other existing robust recourse generating methods from the literature, using both $\ell_1$ and PFC costs. Table 3 presents the results for $\ell_1$ cost, while the results for PFC cost are provided in Table 4. In our experiments, we assume that the invalidation targets specified by the user are 0.10 and 0.05 for PiRR. The results show that PiRR generates more robust recourse solutions that meet the user's invalidation targets compared to existing baselines. Moreover, PiRR generates solutions that are easier to implement compared to other methods. Additional comparisons between PiRR and other methods, as well as the impact of $\epsilon$ on the performance of PiRR, can be found in Appendix C.

## 7 Conclusion

In this paper, we have defined a recourse invalidation rate under model shifts and proposed an uncertainty quantification method to provide theoretical upper-bounds on the invalidation rate for a given recourse generated by any state-of-the-art algorithm. To generate recourses with different levels of robustness according to users' needs, a nonconvex constrained simulation-based optimization problem has been formulated and an iterative approach has been proposed to solve the optimization problem. Numerical results have been provided to illustrate the derived theoretical bounds on the recourse invalidation rate and show the efficacy of the proposed algorithms on multiple real datasets. However, our proposed method and theoretical results

Table 4: PiRR compared to existing robust recourse generating methods (LR model, PFC cost).

| Dataset | Algorithm | Invalidation rate before shift ($\mathcal{M}_1$) | Invalidation rate after shift ($\mathcal{M}_2$) | Average cost |
|---|---|---|---|---|
| | ROAR | $0.00 \pm 0.00$ | $0.02 \pm 0.01$ | $0.44 \pm 0.12$ |
| | ARAR | $0.00 \pm 0.00$ | $0.02 \pm 0.02$ | $0.36 \pm 0.10$ |
| Criminal justice | PROBE | $0.00 \pm 0.00$ | $0.02 \pm 0.01$ | $\mathbf{0.25 \pm 0.09}$ |
| | DiRRAc | $0.00 \pm 0.00$ | $\mathbf{0.01 \pm 0.02}$ | $0.28 \pm 0.12$ |
| | PiRR(0.10) | $\mathbf{0.00 \pm 0.00}$ | $0.07 \pm 0.02$ | $0.16 \pm 0.06$ |
| | PiRR(0.05) | $\mathbf{0.00 \pm 0.00}$ | $0.03 \pm 0.01$ | $\mathbf{0.25 \pm 0.08}$ |
| | ROAR | $0.00 \pm 0.00$ | $0.09 \pm 0.07$ | $1.20 \pm 0.10$ |
| | ARAR | $0.00 \pm 0.00$ | $0.06 \pm 0.07$ | $0.92 \pm 0.09$ |
| Student performance | PROBE | $0.00 \pm 0.00$ | $0.04 \pm 0.07$ | $0.74 \pm 0.10$ |
| | DiRRAc | $0.00 \pm 0.00$ | $0.04 \pm 0.06$ | $0.81 \pm 0.08$ |
| | PiRR(0.10) | $\mathbf{0.00 \pm 0.00}$ | $0.07 \pm 0.02$ | $\mathbf{0.85 \pm 0.08}$ |
| | PiRR(0.05) | $\mathbf{0.00 \pm 0.00}$ | $\mathbf{0.03 \pm 0.02}$ | $\mathbf{0.72 \pm 0.10}$ |
| | ROAR | $0.00 \pm 0.00$ | $\mathbf{0.00 \pm 0.00}$ | $0.36 \pm 0.08$ |
| | ARAR | $0.00 \pm 0.00$ | $0.02 \pm 0.02$ | $0.27 \pm 0.06$ |
| German credit | PROBE | $0.00 \pm 0.00$ | $0.02 \pm 0.01$ | $\mathbf{0.27 \pm 0.07}$ |
| | DiRRAc | $0.00 \pm 0.00$ | $0.01 \pm 0.02$ | $0.32 \pm 0.08$ |
| | PiRR(0.10) | $\mathbf{0.00 \pm 0.00}$ | $0.07 \pm 0.02$ | $0.21 \pm 0.06$ |
| | PiRR(0.05) | $\mathbf{0.00 \pm 0.00}$ | $\mathbf{0.02 \pm 0.02}$ | $\mathbf{0.26 \pm 0.07}$ |

are limited to model shifts that satisfy the conditions described in Assumption 2, and the derived theoretical bounds are not strictly tight. In the future, we aim to generalize the proposed method to a wider range of model shifts and further refine our results to achieve tighter bounds.

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

# A Preliminaries

## A.1 Background on algorithm recourse

Suppose $f : \mathcal{X} \to \mathcal{Y}$ is a classifier that maps features $\boldsymbol{x} \in \mathcal{X} \subset \mathbb{R}^d$ to labels $\mathcal{Y} = \{0, 1\}$, where 0 is the unfavorable outcome and 1 is the favorable outcome. Define $f(\boldsymbol{x}) = g(h(\boldsymbol{x}))$, where $h$ is the scoring function and the activation function is $g(t) = \boldsymbol{I}_{t \geq \eta}$. For ease of illustration, we adopt the setting of loan approval as an example, i.e., $h(\boldsymbol{x}) \geq \eta$ denotes that a loan is granted and $h(\boldsymbol{x}) < \eta$ denotes that it is denied. For an individual $\boldsymbol{x}_0$ that was denied by the loan-granting institution, counterfactual explanation methods could provide the individual with a recourse by identifying which attributes to change for reversing the unfavorable prediction result. Given a cost function $c : \mathbb{R}^d \to \mathbb{R}_+$, the counterfactual explanation $\boldsymbol{x}^{\mathrm{CF}}$ can be found by solving (Wachter et al., 2017; Ustun et al., 2019)

$$\min_{\boldsymbol{x}' \in \mathcal{A}} l(f(\boldsymbol{x}'), 1) + \lambda c(\boldsymbol{x}, \boldsymbol{x}'), \tag{3}$$

where $\mathcal{A}$ is the set of actionable counterfactuals, $\lambda$ is the trade-off parameter, and $l$ is the loss for invalid recourse. The first term in the objective function guarantees that the prediction result of the counterfactual $\boldsymbol{x}'$ is close to the favorable outcome 1. The second term in the objective function encourages the recourse to have lower cost.

## A.2 Background on conformal predictive inference

Conformal prediction is a machine learning framework designed for quantifying uncertainty, extensively explored in (Vovk et al., 2005). More recently, the conformal inference framework has evolved to offer a methodology for converting the outputs of any black-box prediction algorithm into a prediction set (Gibbs & Candes, 2021). The algorithms from conformal inference provide a prediction set that has valid marginal coverage $\mathbb{P}(Y_i \in \hat{C}(\boldsymbol{X}_i)) \geq 1 - \alpha$ based on standard properties of quantiles, if the training and test data are exchangeable (Cauchois et al., 2020; Gibbs & Candes, 2021).

To produce the prediction set, a conformal predictor uses a nonconformity function, an arbitrary function $s : \mathcal{X} \times \mathcal{Y} \to \mathbb{R}$, that measures the strangeness of a sample $(\boldsymbol{x}, y)$ (Johansson et al., 2017). Based on the nonconformity scores of examples with known output labels, and the nonconformity score of a tentatively labels test pattern $(\boldsymbol{x}_{n+1}, \tilde{y})$, a $p$-value statistic can be calculated to reject the hypothesis that $\tilde{y}$ corresponds with the true label $y_{n+1}$. Then all labels $\tilde{y} \subset \mathcal{Y}$ that are not rejected at the chosen significance level $\alpha$ constitute the final prediction set, which contains the true label $y_{n+1}$ with a probability of $1 - \alpha$.

In particular, for a given confidence level $(1 - \alpha)$, one can define a confidence set $\hat{C}(\boldsymbol{x})$ based on the validation set $\mathcal{D}_{\mathrm{val}} = \{(\boldsymbol{X}_i, Y_i)\}_{i=1}^n$, i.e.

$$\hat{C}(\boldsymbol{x}) = \{y \in \mathcal{Y} | s(\boldsymbol{x}, y) \leq \hat{\mathcal{Q}}_{n, 1-\alpha}\}, \tag{4}$$

where

$$\hat{\mathcal{Q}}_{n, 1-\alpha} = \mathrm{Quantile}\left(\left(1 + \frac{1}{n}\right)\alpha; \{s(\boldsymbol{X}_i, Y_i)\}_{i=1}^n\right).$$

Then as long as $\{(\boldsymbol{X}_i, Y_i)\}_{i=1}^{n+1}$ are exchangeable, the confidence set $\hat{C}(\boldsymbol{X}_{n+1})$ satisfies (Romano et al., 2019)

$$\mathbb{P}(Y_{n+1} \in \hat{C}(\boldsymbol{X}_{n+1})) \geq 1 - \alpha. \tag{5}$$

In the algorithmic recourse scenario, we view the counterfactual sample $(\boldsymbol{x}^{\mathrm{CF}}, y^{\mathrm{CF}})$ as the $(n + 1)$-th test sample. Then the nonconformity score $s(\boldsymbol{x}^{\mathrm{CF}}, y^{\mathrm{CF}})$ measures the degree of nonconformity between the counterfactual sample and samples in $\mathcal{D}$. Different from the above-mentioned inference problem, we do not have a prescribed value of $\alpha$, but have some observed properties on the value of $s(\boldsymbol{x}^{\mathrm{CF}}, y^{\mathrm{CF}})$. Thus, by transforming equation 4, equation 5 and applying them to $(\boldsymbol{x}^{\mathrm{CF}}, y^{\mathrm{CF}})$, we have

$$\mathbb{P}(s(\boldsymbol{x}^{\mathrm{CF}}, y^{\mathrm{CF}}) \leq \hat{\mathcal{Q}}_{n, 1-\alpha}) \geq 1 - \alpha, \tag{6}$$

where the value of $\alpha$ can be derived based on the known properties of $s(\boldsymbol{x}^{\mathrm{CF}}, y^{\mathrm{CF}})$. Moreover, equation 6 provides a probability inequality on the value of $s(\boldsymbol{x}^{\mathrm{CF}}, y^{\mathrm{CF}})$, which is useful in measuring the robustness of $\boldsymbol{x}^{\mathrm{CF}}$.

However, the above mentioned results are limited by the exchangeable data assumption. Recently, there are works extending the conformal inference beyond the case of exchangeable data. In particular, a weighted version of conformal inference has been proposed to compute distribution-free prediction intervals for problems in which the test and training covariant distributions differ, but the likelihood ratio between the two distributions is known (Tibshirani et al., 2019).

Assume that $\{(\boldsymbol{X}_i, Y_i)\}_{i=1}^n \overset{i.i.d.}{\sim} \mathcal{P}$ and the independent test sample $(\boldsymbol{X}_{n+1}, Y_{n+1}) \sim \mathcal{P}'$. Then the likelihood ratio between $\mathcal{P}$ and $\mathcal{P}'$ is defined as

$$v(\boldsymbol{x}, y) = \frac{d\mathcal{P}'}{d\mathcal{P}}(\boldsymbol{x}, y), \tag{7}$$

and $v(\boldsymbol{x}, y)$ is assumed to be known exactly in (Tibshirani et al., 2019). For any new data sample $(\boldsymbol{x}, y) \in \mathcal{X} \times \mathcal{Y}$ (e.g. the generated counterfactual sample $(\boldsymbol{x}^{\mathrm{CF}}, y^{\mathrm{CF}})$), assign weights to the sample as

$$
\begin{aligned}
p_i(\boldsymbol{x}, y) &= \frac{v(\boldsymbol{X}_i, Y_i)}{\sum_{j=1}^n v(\boldsymbol{X}_j, Y_j) + v(\boldsymbol{x}, y)}, i = 1, 2, \cdots, n, \\
p_{n+1}(\boldsymbol{x}, y) &= \frac{v(\boldsymbol{x}, y)}{\sum_{j=1}^n v(\boldsymbol{X}_j, Y_j) + v(\boldsymbol{x}, y)}.
\end{aligned}
\tag{8}
$$

Then we have

$$\mathbb{P}'(Y_{n+1} \in \hat{D}(\boldsymbol{X}_{n+1})) \geq 1 - \alpha,$$

where the prediction interval $\hat{D}(\boldsymbol{X}_{n+1})$ is given by

$$\hat{D}(\boldsymbol{X}_{n+1}) = \{y : s(\boldsymbol{X}_{n+1}, y) \leq \hat{S}_{1-\alpha}(y)\},$$

with

$$\hat{S}_{1-\alpha}(y) = \mathrm{Quantile}\left(1 - \alpha, \sum_{i=1}^n p_i(\boldsymbol{X}_{n+1}, y)\delta_{v_{n+1}(S_i)} + p_{n+1}(\boldsymbol{X}_{n+1}, y)\delta_\infty\right),$$

and $\delta$ denotes the point mass.

In the algorithm recourse scenario, since the distribution of nonconformity score variable $s$ changes, we can leverage the concept of weighted conformal inference by assigning weights to the validation samples based on their similarity to the counterfactual example to provide probability bounds on the nonconformity score of $(\boldsymbol{x}^{\mathrm{CF}}, y^{\mathrm{CF}})$.

# B Additional Algorithms

## B.1 Algorithm for finding $k^*$ given a target level $\alpha$

---

**Algorithm 3** Procedure of finding $k^*$ given a target level $\alpha$

---

**Input:** training data $\mathcal{D}_{\text{train}}$, calibration data $\mathcal{D}_{\text{calib}}$, recourse $\boldsymbol{x}^{\text{CF}}$, nonconformity score function $s(\cdot, \cdot)$, model shift parameter $\tau$, target level $\alpha \in (0, 1)$.

1: Use $\mathcal{D}_{\text{train}}$ to construct the point-wise bounds $\hat{L}(\cdot)$ and $\hat{U}(\cdot)$ for likelihood ratio $v$.
2: For samples in $\mathcal{D}_{\text{calib}}$, compute $S_i = s(\boldsymbol{X}_i, Y_i), L_i = \hat{L}(S_i), U_i = \hat{U}(S_i)$.
3: For the recourse $\boldsymbol{x}^{\text{CF}}$, compute $L^{\text{CF}}$ and $U^{\text{CF}}$.
4: For $1 \leq k \leq n$, compute $\hat{F}(k)$.
5: Derive $k^* = \min\{k : \hat{F}(k) \geq 1 - \alpha\}$.

**Output:** The value of $k^*$.

---

## B.2 Recourses with various choices of $\lambda$

In this subsection, we provide an additional algorithm for finding recourses with different robustness levels. The performance of this algorithm can be found in Figure 5.

In state-of-the-art recourse algorithms (Wachter et al., 2017; Ustun et al., 2019; Karimi et al., 2020a), the low-cost recourse for an adversely predicted sample $\boldsymbol{x}_0$ is found by solving

$$\boldsymbol{x}^{\text{CF}} = \arg\min_{\boldsymbol{x} \in \mathcal{A}}[l(f(\boldsymbol{x}), 1) + \lambda c(\boldsymbol{x}_0, \boldsymbol{x})], \tag{9}$$

where the trade-off parameter $\lambda$ is considered given. However, we note that as the value of $\lambda$ changes, the generated recourse $\boldsymbol{x}^{\text{CF}}$ varies, and the recourse cost as well as the recourse invalidation rate also change accordingly. Thus, a natural way to find different recourses with different robustness levels is to vary the value of $\lambda$. For each choice of $\lambda$ (e.g. $\lambda = \lambda_j$), we can generate a recourse $\boldsymbol{x}_j^{\text{CF}}$ by any recourse generating algorithm and derive the corresponding recourse cost $c_j = c(\boldsymbol{x}_0, \boldsymbol{x}_j^{\text{CF}})$. Based on Theorem 9 or Theorem 10, we are able to derive the upper-bound $r_{u,j} = r_u(\boldsymbol{x}_j^{\text{CF}})$ on the recourse invalidation rate, which measures the robustness of $\boldsymbol{x}_j^{\text{CF}}$ to model changes. Then all generated recourses and their corresponding recourse costs as well as robustness metrics (bounds on the recourse invalidation rates), i.e. $\{(\boldsymbol{x}_j^{\text{CF}}, c_j, r_{u,j})\}_j$, can be provided to users. We summarize the procedure in Algorithm 4.

---

**Algorithm 4** Recourses with various choices of $\lambda$

---

**Input:** negatively predicted sample $\boldsymbol{x}_0$, current model $f$, model shift parameter $\tau$, maximum trade-off parameter $\lambda_m$, increment parameter $d_\lambda$ of $\lambda$.

1: $j = 1$
2: **for** $\lambda = 0 : d_\lambda : \lambda_m$ **do**
3:   solve equation 9 by a recourse generating algorithm and the generated recourse is $\boldsymbol{x}_j^{\text{CF}}$;
4:   calculate the recourse cost $c_j = c(\boldsymbol{x}_0, \boldsymbol{x}_j^{\text{CF}})$;
5:   derive the upper-bound $r_{u,j} = r_u(\boldsymbol{x}_j^{\text{CF}})$ (according to Theorems 9 or 10) on the invalidation rate;
6:   $j = j + 1$;
7: **end for**

**Output:** $\{(\boldsymbol{x}_j^{\text{CF}}, c_j, r_{u,j})\}_j$.

---

## C   Additional Details on Numerical Results

All experiments were run on a 2.8 GHz Quad-Core Intel Core i7.

### C.1   Details about the datasets

We conduct our analysis using three real datasets: Criminal justice dataset (Lakkaraju et al., 2016), Student performance dataset (Amrieh et al., 2016) and German credit dataset (Dua & Graff, 2017). Each dataset contains two parts, initial data ($D_1$) and shifted data ($D_2$).

1. Criminal justice dataset (Lakkaraju et al., 2016): It contains proprietary data from 1978 ($D_1$) and 1980 ($D_2$), with 8395 and 8595 samples, respectively. It includes demographic features such as race, sex, age, time-served, and employment, and a target attribute related to bail decisions. Furthermore, the dataset exhibits an inherent temporal shift, as the data characteristics in 1980 differ from those in 1978.

2. Student performance dataset (Amrieh et al., 2016): It comprises publicly available data collected from schools in Jordan ($D_1$) and Kuwait ($D_2$), with 129 and 122 samples, respectively. The problem of predicting grades is viewed as a binary classification task, with numerical grades transformed into pass and fail. Predictors such as grade, holidays-taken, and class-participation are included, and the dataset demonstrates an inherent geospatial distribution shift as the data characteristics of students vary across countries. The features we use are: "sex", "age", "address", "famsize", "Pstatus", "Medu", "Fedu", "Mjob", "Fjob", "reason", "guardian", "traveltime", "studytime", "failures", "schoolsup", "famsup", "paid", "activities", "nursery", "higher", "internet", "romantic", "famrel", "freetime", "goout", "Dalc", "Walc", "health", "absences".

3. German credit dataset (Dua & Graff, 2017): It contains 900 samples from two versions each. The applicants' loan amount, employment history, and age are used to predict their credit score. Additionally, the data exhibits a data correction-based distribution shift, as the data's characteristics differ due to a change in the data preprocessing step. The features we use are: "duration", "amount", "age", "personal-status-sex".

### C.2   Classification models

This subsection outlines the fitting process for the classification models. A standard 4 : 1 train-test split was employed for model training and evaluation. Identical architectures were used for all models across the datasets, as shown in Table 5. The model performance is evaluated based on the accuracy as shown in Table 6.

Table 5: Classification models architecture

|  | LR | NN |
|---|---|---|
| Units | [Input dimension, 2] | [Input dimension, 50, 2] |
| Type | Fully connected | Fully connected |
| Intermediate activation | NA | ReLu |
| Last layer activation | Softmax | Softmax |

Table 6: Average test accuracy for classification models

|  | Criminal justice | Student performance | German credit |
|---|---|---|---|
| LR | $1.00 \pm 0.00$ | $0.92 \pm 0.01$ | $0.70 \pm 0.01$ |
| NN | $1.00 \pm 0.00$ | $0.95 \pm 0.01$ | $0.75 \pm 0.02$ |

## C.3 Implementation details

For a given dataset, a particular predictive model (NN or LR), and a specific baseline recourse generating method, to validate the theoretical bounds on the recourse invalidation rate, we

1. train predictive model $\mathcal{M}_1$ on the training fold of $D_1$;

2. use $\mathcal{M}_1$ to obtain prediction result for each sample in the validation fold of $D_1$;

3. select samples that have negative prediction results;

4. generate recourses for those negatively-predicted samples based on $\mathcal{M}_1$ by using the specified recourse generating method;

5. derive the updated model $\mathcal{M}_2$ on the shifted data $D_2$;

6. verify Assumption 2 and derive the value of $\tau$ based on $\mathcal{M}_2$;

7. for each recourse, compute bounds on the recourse invalidation rate according to Theorems 9 and 10 (since the bounds are also derived through simulation, we need to run Algorithm 1 when computing the bounds);

8. use $\mathcal{M}_2$ to obtain prediction result for each recourse and evaluate the empirical recourse invalidation rate;

9. compare the empirical invalidation rate and the theoretical bounds.

## C.4 Additional experimental results

In Table 7 and Table 8, we provide empirical invalidation rates of recourses generated by baseline algorithms. We report the averaged empirical invalidation rate as well as its standard deviation.

Table 7: Empirical invalidation rate of recourses under model shifts ($\ell_1$ cost)

| Algorithm | Dataset | Predictive model | Empirical invalidation rate |
|---|---|---|---|
| CF | Criminal justice | LR | $0.69 \pm 0.09$ |
| | | NN | $0.48 \pm 0.09$ |
| | Student performance | LR | $0.71 \pm 0.09$ |
| | | NN | $0.52 \pm 0.09$ |
| | German credit | LR | $0.46 \pm 0.27$ |
| | | NN | $0.53 \pm 0.06$ |
| AR | Criminal justice | LR | $0.84 \pm 0.06$ |
| | | NN | $0.35 \pm 0.17$ |
| | Student performance | LR | $0.57 \pm 0.14$ |
| | | NN | $0.17 \pm 0.10$ |
| | German credit | LR | $0.47 \pm 0.21$ |
| | | NN | $0.69 \pm 0.06$ |
| MINT | German credit | LR | $0.07 \pm 0.07$ |
| | | NN | $0.37 \pm 0.11$ |

Figure 3 compares the performance of PiRR with baseline methods in generating recourse under 3 different prescribed invalidation rates: $0.05, 0.10, 0.15$. Figure 4 investigates the impact of $\epsilon$ on the performance of PiRR.

For Algorithm 4, we use the considered three baseline recourse generating methods to generate recourses for negatively-predicted samples in the validation fold. To obtain recourses with different costs and robustness, we vary the value of the trade-off parameter $\lambda$. In particular, we choose $\lambda = \{0.1, 0.5, 0.9, 1.3, 1.7\}$. The results are shown in Figure 5.

Table 8: Empirical invalidation rate of recourses under model shifts (PFC cost)

| Algorithm | Dataset | Predictive model | Empirical invalidation rate |
|---|---|---|---|
| CF | Criminal justice | LR | $0.74 \pm 0.11$ |
| | | NN | $0.50 \pm 0.13$ |
| | Student performance | LR | $0.82 \pm 0.10$ |
| | | NN | $0.70 \pm 0.14$ |
| | German credit | LR | $0.44 \pm 0.33$ |
| | | NN | $0.49 \pm 0.12$ |
| AR | Criminal justice | LR | $0.91 \pm 0.05$ |
| | | NN | $0.65 \pm 0.17$ |
| | Student performance | LR | $0.76 \pm 0.11$ |
| | | NN | $0.18 \pm 0.11$ |
| | German credit | LR | $0.46 \pm 0.27$ |
| | | NN | $0.44 \pm 0.15$ |
| MINT | German credit | LR | $0.05 \pm 0.08$ |
| | | NN | $0.36 \pm 0.15$ |

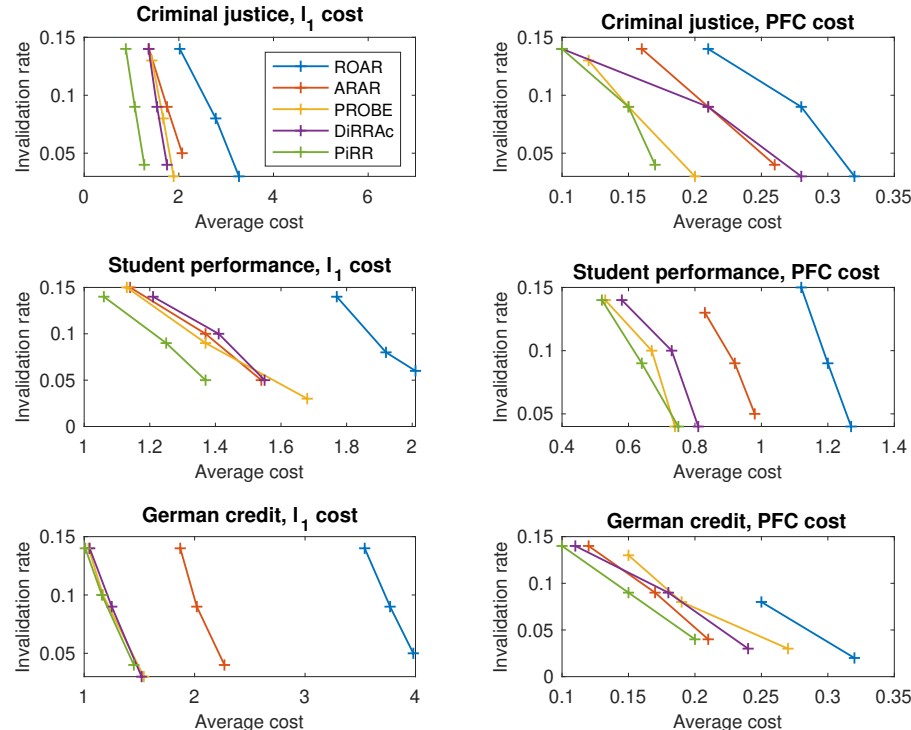

Figure 3: Recourse invalidation rate v.s. recourse cost plot. For any given invalidation rate, PiRR could generate recourses that satisfy the invalidation requirement while maintaining low recourse costs. The average recourse costs of robust recourses generated by PiRR are smaller than other methods under the same invalidation rate constraint.

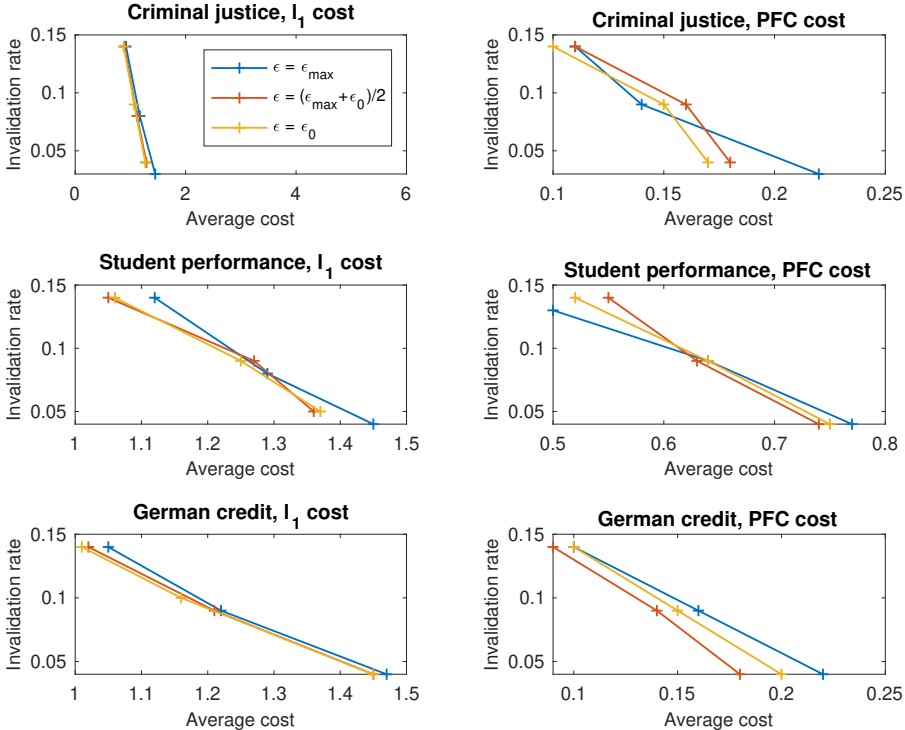

Figure 4: Impact of $\epsilon$ on the performance of PiRR, where $\epsilon_0 = \min\limits_{\{s \in \mathcal{S}_{\text{train}}\}} \hat{p}(s), \epsilon_{\max} = \sum\limits_{\{s' \in \mathcal{S}_{\text{train}}: s-\tau \leq s' \leq s+\tau\}} \hat{p}(s')$. PiRR consistently generates recourses that meet the invalidation requirements across different values of $\epsilon$, while maintaining similar overall performance.

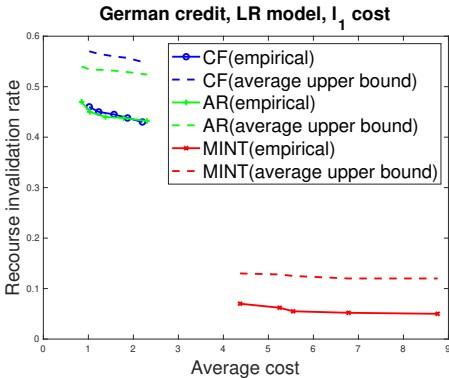

Figure 5: Recourse invalidation rate v.s. recourse cost plot for Algorithm 4. As $\lambda$ varies, the recourse cost changes, while the recourse invalidation rate changes only slightly. The theoretical bounds on the recourse invalidation rates are valid.

# D Proofs

## D.1 Proof of Lemma 4

$$
\begin{aligned}
r_{\mathrm{ivd}}(\boldsymbol{x}^{\mathrm{CF}}) &= \mathbb{P}(f'(\boldsymbol{x}^{\mathrm{CF}}) = 0 | f(\boldsymbol{x}^{\mathrm{CF}}) = 1) \\
&= 1 - \mathbb{P}(h'(\boldsymbol{x}^{\mathrm{CF}}) \geq \eta | h(\boldsymbol{x}^{\mathrm{CF}}) \geq \eta) \\
&\overset{(a)}{=} 1 - \mathbb{P}(h'(\boldsymbol{x}^{\mathrm{CF}}) \geq \eta | h(\boldsymbol{x}^{\mathrm{CF}}) \geq \eta, h'(\boldsymbol{x}^{\mathrm{CF}}) \geq \eta - \tau) \\
&= 1 - \frac{\mathbb{P}(h'(\boldsymbol{x}^{\mathrm{CF}}) \geq \eta, h(\boldsymbol{x}^{\mathrm{CF}}) \geq \eta | h'(\boldsymbol{x}^{\mathrm{CF}}) \geq \eta - \tau)}{\mathbb{P}(h(\boldsymbol{x}^{\mathrm{CF}}) \geq \eta | h'(\boldsymbol{x}^{\mathrm{CF}}) \geq \eta - \tau)} \\
&= 1 - \frac{\mathbb{P}(h'(\boldsymbol{x}^{\mathrm{CF}}) \geq \eta | h'(\boldsymbol{x}^{\mathrm{CF}}) \geq \eta - \tau)\mathbb{P}(h(\boldsymbol{x}^{\mathrm{CF}}) \geq \eta | h'(\boldsymbol{x}^{\mathrm{CF}}) \geq \eta, h'(\boldsymbol{x}^{\mathrm{CF}}) \geq \eta - \tau)}{\mathbb{P}(h(\boldsymbol{x}^{\mathrm{CF}}) \geq \eta | h'(\boldsymbol{x}^{\mathrm{CF}}) \geq \eta - \tau)} \\
&= 1 - \mathbb{P}(h'(\boldsymbol{x}^{\mathrm{CF}}) \geq \eta | h'(\boldsymbol{x}^{\mathrm{CF}}) \geq \eta - \tau)\frac{\mathbb{P}(h(\boldsymbol{x}^{\mathrm{CF}}) \geq \eta | h'(\boldsymbol{x}^{\mathrm{CF}}) \geq \eta, h'(\boldsymbol{x}^{\mathrm{CF}}) \geq \eta - \tau)}{\mathbb{P}(h(\boldsymbol{x}^{\mathrm{CF}}) \geq \eta | h'(\boldsymbol{x}^{\mathrm{CF}}) \geq \eta - \tau)} \\
&\overset{(b)}{=} 1 - \mathbb{P}(h'(\boldsymbol{x}^{\mathrm{CF}}) \geq \eta | h'(\boldsymbol{x}^{\mathrm{CF}}) \geq \eta - \tau) \\
&\quad \cdot \frac{\mathbb{P}(h(\boldsymbol{x}^{\mathrm{CF}}) \geq \eta | h'(\boldsymbol{x}^{\mathrm{CF}}) \geq \eta, h(\boldsymbol{x}^{\mathrm{CF}}) \geq \eta - \tau, h'(\boldsymbol{x}^{\mathrm{CF}}) \geq \eta - \tau)}{\mathbb{P}(h(\boldsymbol{x}^{\mathrm{CF}}) \geq \eta | h'(\boldsymbol{x}^{\mathrm{CF}}) \geq \eta - \tau)}
\end{aligned}
$$

$$
\begin{aligned}
&\leq 1 - \mathbb{P}(h'(\boldsymbol{x}^{\mathrm{CF}}) \geq \eta | h'(\boldsymbol{x}^{\mathrm{CF}}) \geq \eta - \tau) \\
&= 1 - \frac{\mathbb{P}(h'(\boldsymbol{x}^{\mathrm{CF}}) \geq \eta, h'(\boldsymbol{x}^{\mathrm{CF}}) \geq \eta - \tau)}{\mathbb{P}(h'(\boldsymbol{x}^{\mathrm{CF}}) \geq \eta - \tau)} \\
&= 1 - \frac{\mathbb{P}(h'(\boldsymbol{x}^{\mathrm{CF}}) \geq \eta)}{\mathbb{P}(h'(\boldsymbol{x}^{\mathrm{CF}}) \geq \eta - \tau)} \\
&= 1 - \frac{\mathbb{P}(s'(\boldsymbol{x}^{\mathrm{CF}}, 1) \leq 1 - \eta)}{\mathbb{P}(s'(\boldsymbol{x}^{\mathrm{CF}}, 1) \leq 1 - \eta + \tau)},
\end{aligned}
\tag{10}
$$

where (a) is true as $h(\boldsymbol{x}^{\mathrm{CF}}) \geq \eta$ implies $h'(\boldsymbol{x}^{\mathrm{CF}}) \geq \eta - \tau$ based on Assumption 2. Similarly, (b) holds because $h'(\boldsymbol{x}^{\mathrm{CF}}) \geq \eta$ implies $h(\boldsymbol{x}^{\mathrm{CF}}) \geq \eta - \tau$ based on Assumption 2.

## D.2 Proof of Lemma 6

Under Assumption 2, we have $h(\boldsymbol{x}) - \tau \leq h'(\boldsymbol{x}) \leq h(\boldsymbol{x}) + \tau, \forall \boldsymbol{x} \in \mathcal{X}$, which implies

$$
s(\boldsymbol{x}, y = 0) - \tau \leq s'(\boldsymbol{x}, y = 0) \leq s(\boldsymbol{x}, y = 0) + \tau,
$$

as well as

$$
s(\boldsymbol{x}, y = 1) - \tau = 1 - h(\boldsymbol{x}) - \tau \leq 1 - h'(\boldsymbol{x}) = s'(\boldsymbol{x}, y = 1) \leq 1 - h(\boldsymbol{x}) + \tau = s(\boldsymbol{x}, y = 1) + \tau.
$$

Then we have

$$
s(\boldsymbol{x}, y) - \tau \leq s'(\boldsymbol{x}, y) \leq s(\boldsymbol{x}, y) + \tau,
$$

which indicates that to derive $p'$ from $p$, only density in the $\tau$-neighborhood of $s$ can be moved to $s$. Then for any neighborhood of $s$ with radius $\delta$, the cumulative probability under the distribution $p'$ over this neighborhood is always upper-bounded by the cumulative probability under the distribution $p$ over a larger neighborhood of radius $\tau + \delta$ around $s$. Specifically, we have

$$
\int_{s-\delta}^{s+\delta} p'(t)dt \leq \int_{s-\tau-\delta}^{s+\tau+\delta} p(t)d(t).
$$

### D.3 Proof of Proposition 7

In the following, we denote the random variables $S_i$ and realized values $s_i, i = 1, 2, \cdots, n$. For notation simplicity, we denote $s_{n+1} = s'(\boldsymbol{x}^{\mathrm{CF}}, y^{\mathrm{CF}})$. From (Tibshirani et al., 2019), we know that independent draws are always weighted exchangeable, with weight functions given by likelihood ratios. Thus, according to Definition 1 and Lemma 2 in (Tibshirani et al., 2019), we have that random variables $S_1, \cdots, S_{n+1}$ are weighted exchangeable and

$$f(s_1, \cdots, s_{n+1}) = \Pi_{i=1}^{n+1} v_i(s_i) g(s_1, \cdots, s_{n+1}), \tag{11}$$

where $f$ represents the joint pdf, $v_i(s_i) = 1, i = 1, \cdots n, v_{n+1}(s_{n+1}) = v(s'(\boldsymbol{x}^{\mathrm{CF}}, y^{\mathrm{CF}}))$ and $g$ is a permutation-invariant function.

For a set of values $s_1, \cdots, s_{n+1}$ where there might be repeated elements, we denote the unordered set $\boldsymbol{s} = [s_1, \cdots, s_{n+1}]$ and denote an event $E_{\boldsymbol{s}} = \{[S_1, S_2, \cdots, S_{n+1}] = [s_1, s_2, \cdots, s_{n+1}]\}$. Let $\Pi_{n+1}$ be the set of all permutations of $\{1, \cdots, n+1\}$. Then we have

$$
\begin{aligned}
&\mathbb{P}(S_{n+1} = s_i | E_{\boldsymbol{s}}) \\
&= \frac{\sum\limits_{\pi \in \Pi_{n+1} : \pi(n+1) = i} f(s_{\pi(1)}, s_{\pi(2)}, \cdots, s_{\pi(n+1)})}{\sum\limits_{\pi \in \Pi_{n+1}} f(s_{\pi(1)}, s_{\pi(2)}, \cdots, s_{\pi(n+1)})} \\
&\overset{(a)}{=} \frac{\sum\limits_{\pi \in \Pi_{n+1} : \pi(n+1) = i} \Pi_{i=1}^{n+1} v_i(s_{\pi(i)}) g(s_{\pi(1)}, s_{\pi(2)}, \cdots, s_{\pi(n+1)})}{\sum\limits_{\pi \in \Pi_{n+1}} \Pi_{i=1}^{n+1} v_i(s_{\pi(i)}) g(s_{\pi(1)}, s_{\pi(2)}, \cdots, s_{\pi(n+1)})} \\
&= \frac{\sum\limits_{\pi \in \Pi_{n+1} : \pi(n+1) = i} v_{n+1}(s_i) g(s_{\pi(1)}, s_{\pi(2)}, \cdots, s_{\pi(n+1)})}{\sum\limits_{\pi \in \Pi_{n+1}} v_{n+1}(s_{\pi(n+1)}) g(s_{\pi(1)}, s_{\pi(2)}, \cdots, s_{\pi(n+1)})} \\
&= \frac{v_{n+1}(s_i)}{\sum\limits_{j=1}^{n+1} v_{n+1}(s_j)} = \frac{v(s_i)}{\sum\limits_{j=1}^{n+1} v(s_j)} := p_i,
\end{aligned}
$$

where (a) is due to equation 11.

Then for any unordered set $\boldsymbol{s}$, we have

$$\mathbb{P}(S_{n+1} \leq s_{[k^*]} | E_{\boldsymbol{s}}) = \sum_{i=1}^{n+1} p_i \mathbf{1}_{s_i \leq s_{[k^*]}} = \frac{\sum\limits_{i=1}^{n+1} v(s_i) \mathbf{1}_{s_i \leq s_{[k^*]}}}{\sum\limits_{j=1}^{n+1} v(s_j)}. \tag{12}$$

Recall that $\hat{F}$ is defined as

$$\hat{F}(k) = \frac{\sum\limits_{i=1}^{k} L_{[i]}}{\sum\limits_{i=1}^{k} L_{[i]} + \sum\limits_{i=k+1}^{n} U_{[i]} + U^{\mathrm{CF}}}.$$

Since $k^* = \min\{k : \hat{F}(k) \geq 1 - \alpha\}$, we have

$$S_{[k^*]} = \inf \left\{ s : \frac{\sum\limits_{i=1}^{n} L_i \mathbf{1}_{S_i \leq s}}{\sum\limits_{i=1}^{n} L_i \mathbf{1}_{S_i \leq s} + \sum\limits_{i=1}^{n} U_i \mathbf{1}_{S_i > s} + U^{\mathrm{CF}}} \geq 1 - \alpha \right\},$$

which indicates that

$$\mathbb{E}\left[\frac{\sum\limits_{i=1}^{n} L_i \mathbf{1}_{S_i \leq S_{[k^*]}}}{\sum\limits_{i=1}^{n} L_i \mathbf{1}_{S_i \leq S_{[k^*]}} + \sum\limits_{i=1}^{n} U_i \mathbf{1}_{S_i > S_{[k^*]}} + U^{\mathrm{CF}}}\right] \geq 1 - \alpha. \tag{13}$$

In the meantime, we apply the power property on equation 12 and have

$$\mathbb{P}(S_{n+1} \leq s_{[k^*]}) = \mathbb{E}[\mathbb{P}(S_{n+1} \leq s_{[k^*]}|E_{\boldsymbol{S}})] = \mathbb{E}\left[\frac{\sum\limits_{i=1}^{n+1} v(S_i) \mathbf{1}_{S_i \leq S_{[k^*]}}}{\sum\limits_{j=1}^{n+1} v(S_i)}\right]. \tag{14}$$

By combining equation 13 and equation 14, we have

$$
\begin{aligned}
&\mathbb{P}(S_{n+1} \leq s_{[k^*]}) - (1 - \alpha) \\
&\geq \ \mathbb{E}\left[\frac{\sum\limits_{i=1}^{n+1} v(S_i) \mathbf{1}_{S_i \leq S_{[k^*]}}}{\sum\limits_{i=1}^{n+1} v(S_i)}\right] - \mathbb{E}\left[\frac{\sum\limits_{i=1}^{n} L_i \mathbf{1}_{S_i \leq S_{[k^*]}}}{\sum\limits_{i=1}^{n} L_i \mathbf{1}_{S_i \leq S_{[k^*]}} + \sum\limits_{i=1}^{n} U_i \mathbf{1}_{S_i > S_{[k^*]}} + U^{\mathrm{CF}}}\right] \\
&\geq \ \mathbb{E}\left[\frac{\sum\limits_{i=1}^{n} v(S_i) \mathbf{1}_{S_i \leq S_{[k^*]}}}{\sum\limits_{i=1}^{n+1} v(S_i)}\right] - \mathbb{E}\left[\frac{\sum\limits_{i=1}^{n} L_i \mathbf{1}_{S_i \leq S_{[k^*]}}}{\sum\limits_{i=1}^{n} L_i \mathbf{1}_{S_i \leq S_{[k^*]}} + \sum\limits_{i=1}^{n} U_i \mathbf{1}_{S_i > S_{[k^*]}} + U^{\mathrm{CF}}}\right] \\
&= \ \mathbb{E}\left[\frac{q(S_1, \cdots, S_{n+1})}{\left[\sum\limits_{i=1}^{n+1} v(S_i)\right]\left[\sum\limits_{i=1}^{n} L_i \mathbf{1}_{S_i \leq S_{[k^*]}} + \sum\limits_{i=1}^{n} U_i \mathbf{1}_{S_i > S_{[k^*]}} + U^{\mathrm{CF}}\right]}\right],
\end{aligned} \tag{15}
$$

where

$$
\begin{aligned}
&q(S_1, \cdots, S_{n+1}) \\
&= \ \left[\sum_{i=1}^{n} v(S_i)\mathbf{1}_{S_i \leq S_{[k^*]}}\right] \cdot \left[\sum_{i=1}^{n} L_i \mathbf{1}_{S_i \leq S_{[k^*]}}\right] + \left[\sum_{i=1}^{n} v(S_i)\mathbf{1}_{S_i \leq S_{[k^*]}}\right] \cdot \left[\sum_{i=1}^{n} U_i \mathbf{1}_{S_i > S_{[k^*]}}\right] \\
&\quad + \left[\sum_{i=1}^{n} v(S_i)\mathbf{1}_{S_i \leq S_{[k^*]}}\right] U^{\mathrm{CF}} - \left[\sum_{i=1}^{n+1} v(S_i)\right] \cdot \left[\sum_{i=1}^{n} L_i \mathbf{1}_{S_i \leq S_{[k^*]}}\right] \\
&= \ \left[\sum_{i=1}^{n} v(S_i)\mathbf{1}_{S_i \leq S_{[k^*]}}\right] \cdot \left[\sum_{i=1}^{n} L_i \mathbf{1}_{S_i \leq S_{[k^*]}}\right] - \left[\sum_{i=1}^{n+1} v(S_i)\right] \cdot \left[\sum_{i=1}^{n} L_i \mathbf{1}_{S_i \leq S_{[k^*]}}\right] \\
&\quad + \left[\sum_{i=1}^{n} v(S_i)\mathbf{1}_{S_i \leq S_{[k^*]}}\right] \cdot \left[\sum_{i=1}^{n} U_i \mathbf{1}_{S_i > S_{[k^*]}}\right] + \left[\sum_{i=1}^{n} v(S_i)\mathbf{1}_{S_i \leq S_{[k^*]}}\right] U^{\mathrm{CF}} \\
&= \ \left\{\left[\sum_{i=1}^{n} v(S_i)\mathbf{1}_{S_i \leq S_{[k^*]}}\right] \cdot \left[\sum_{i=1}^{n} U_i \mathbf{1}_{S_i > S_{[k^*]}}\right] - \left[\sum_{i=1}^{n} v(S_i)\mathbf{1}_{S_i > S_{[k^*]}}\right] \cdot \left[\sum_{i=1}^{n} L_i \mathbf{1}_{S_i \leq S_{[k^*]}}\right]\right\} \\
&\quad + \left\{U^{\mathrm{CF}}\left[\sum_{i=1}^{n} v(S_i)\mathbf{1}_{S_i \leq S_{[k^*]}}\right] - v(S_{n+1})\left[\sum_{i=1}^{n} L_i \mathbf{1}_{S_i \leq S_{[k^*]}}\right]\right\}
\end{aligned}
$$

$$\overset{(b)}{\geq} \quad \left\{ \left[ \sum_{i=1}^{n} v(S_i)\mathbf{1}_{S_i \leq S_{[k^*]}} \right] \cdot \left[ \sum_{i=1}^{n} v(S_i)\mathbf{1}_{S_i > S_{[k^*]}} \right] \right.$$

$$\left. - \left[ \sum_{i=1}^{n} v(S_i)\mathbf{1}_{S_i > S_{[k^*]}} \right] \cdot \left[ \sum_{i=1}^{n} (v(S_i) + \max\{0, L_i - v(S_i)\})\mathbf{1}_{S_i \leq S_{[k^*]}} \right] \right\}$$

$$+ \left\{ U^{\mathrm{CF}} \left[ \sum_{i=1}^{n} v(S_i)\mathbf{1}_{S_i \leq S_{[k^*]}} \right] - v(S_{n+1}) \left[ \sum_{i=1}^{n} (v(S_i) + \max\{0, L_i - v(S_i)\})\mathbf{1}_{S_i \leq S_{[k^*]}} \right] \right\}$$

$$= \quad \left\{ \left[ \sum_{i=1}^{n} v(S_i)\mathbf{1}_{S_i \leq S_{[k^*]}} \right] \cdot \left[ \sum_{i=1}^{n} v(S_i)\mathbf{1}_{S_i > S_{[k^*]}} \right] \right.$$

$$- \left[ \sum_{i=1}^{n} v(S_i)\mathbf{1}_{S_i > S_{[k^*]}} \right] \cdot \left[ \sum_{i=1}^{n} v(S_i)\mathbf{1}_{S_i \leq S_{[k^*]}} \right]$$

$$\left. - \left[ \sum_{i=1}^{n} v(S_i)\mathbf{1}_{S_i > S_{[k^*]}} \right] \cdot \left[ \sum_{i=1}^{n} \max\{0, L_i - v(S_i)\}\mathbf{1}_{S_i \leq S_{[k^*]}} \right] \right\}$$

$$+ \left\{ U^{\mathrm{CF}} \left[ \sum_{i=1}^{n} v(S_i)\mathbf{1}_{S_i \leq S_{[k^*]}} \right] - v(S_{n+1}) \left[ \sum_{i=1}^{n} v(S_i)\mathbf{1}_{S_i \leq S_{[k^*]}} \right] \right.$$

$$\left. -v(S_{n+1}) \left[ \sum_{i=1}^{n} \max\{0, L_i - v(S_i)\}\mathbf{1}_{S_i \leq S_{[k^*]}} \right] \right\}$$

$$\geq \quad - \left[ \sum_{i=1}^{n} v(S_i)\mathbf{1}_{S_i > S_{[k^*]}} \right] \cdot \left[ \sum_{i=1}^{n} \max\{0, L_i - v(S_i)\}\mathbf{1}_{S_i \leq S_{[k^*]}} \right]$$

$$-v(S_{n+1}) \left[ \sum_{i=1}^{n} \max\{0, L_i - v(S_i)\}\mathbf{1}_{S_i \leq S_{[k^*]}} \right]$$

$$\geq \quad - \left[ \sum_{i=1}^{n+1} v(S_i) \right] \cdot \left[ \sum_{i=1}^{n} \max\{0, L_i - v(S_i)\} \right], \tag{16}$$

in which (b) is due to the fact that $\hat{U}(s) \geq v(s)$ almost surely since the formula for $\hat{U}(\cdot)$ is motivated by Lemma 6.

Then we come back to equation 15 and have

$$\mathbb{E} \left[ \frac{q(S_1, \cdots, S_{n+1})}{\left[ \sum_{i=1}^{n+1} v(S_i) \right] \left[ \sum_{i=1}^{n} L_i \mathbf{1}_{S_i \leq S_{[k^*]}} + \sum_{i=1}^{n} U_i \mathbf{1}_{S_i > S_{[k^*]}} + U^{\mathrm{CF}} \right]} \right]$$

$$\geq \quad \mathbb{E} \left[ \frac{- \left[ \sum_{i=1}^{n} v(S_i) \right] \cdot \left[ \sum_{i=1}^{n} \max\{0, L_i - v(S_i)\} \right]}{\left[ \sum_{i=1}^{n+1} v(S_i) \right] \left[ \sum_{i=1}^{n} L_i \right]} \right]$$

$$\geq \quad -\mathbb{E} \left[ \frac{\sum_{i=1}^{n} \max\{0, L_i - v(S_i)\}}{\sum_{i=1}^{n} L_i} \right].$$

By Hölder's Inequality, we have

$$\mathbb{E}\left[\frac{\sum_{i=1}^{n} \max\{0, L_i - v(S_i)\}}{\sum_{i=1}^{n} L_i}\right]$$

$$\leq \left\|\frac{1}{n}\sum_{i=1}^{n} \max\{0, L_i - v(S_i)\}\right\|_p \cdot \left\|\frac{n}{\sum_{i=1}^{n} L_i}\right\|_q$$

$$\overset{(b)}{\leq} \|\max\{0, L_i - v(S_i)\}\|_p \cdot \left\|\frac{n}{\sum_{i=1}^{n} L_i}\right\|_q$$

$$\leq \|\max\{0, L_i - v(S_i)\}\|_p \cdot \left\|\frac{1}{n}\sum_{i=1}^{n}\frac{1}{L_i}\right\|_q$$

$$\overset{(c)}{\leq} \|\max\{0, L_i - v(S_i)\}\|_p \cdot \left\|\frac{1}{L_i}\right\|_q,$$

where (b) and (c) follow from the Minkowski's inequality. Thus, we have

$$\mathbb{P}(S_{n+1} \leq s_{[k^*]}) - (1 - \alpha) \geq - \|\max\{0, L_i - v(S_i)\}\|_p \cdot \left\|\frac{1}{L_i}\right\|_q.$$

Since $S_i$ is a random variable and $L_i = \hat{L}(S_i)$, in general, we have

$$\mathbb{P}(S_{n+1} \leq s_{[k^*]}) - (1 - \alpha) \geq - \left\|\frac{1}{\hat{L}(S)}\right\|_q \cdot \left\|\max\{0, \hat{L}(S) - v(S)\}\right\|_p.$$

Proposition 8 can be proved by following similar process.

### D.4 Proof of Theorem 9

Given the definition of $k_1^*$, we have $\mathbb{P}(s'(\boldsymbol{x}^{\mathrm{CF}}, y^{\mathrm{CF}}) \leq s_{[k_1^*]}) \leq \mathbb{P}(s'(\boldsymbol{x}^{\mathrm{CF}}, y^{\mathrm{CF}}) \leq 1 - \eta)$. Then if $s'(\boldsymbol{x}^{\mathrm{CF}}, y^{\mathrm{CF}}) \leq 1 - \eta$, we have

$$1 - \eta \geq s'(\boldsymbol{x}^{\mathrm{CF}}, y^{\mathrm{CF}} = 1) = 1 - h'(\boldsymbol{x}^{\mathrm{CF}})$$

or

$$1 - \eta \geq s'(\boldsymbol{x}^{\mathrm{CF}}, y^{\mathrm{CF}} = 0) = h'(\boldsymbol{x}^{\mathrm{CF}}). \tag{17}$$

However, according to Assumption 2, we have

$$h'(\boldsymbol{x}^{\mathrm{CF}}) \geq \eta - \tau \overset{(a)}{>} 1 - \eta,$$

where (a) is true because $2\eta - \tau > 1$. Then we see a contradiction in equation 17 and conclude that as long as $s'(\boldsymbol{x}^{\mathrm{CF}}, y^{\mathrm{CF}}) \leq 1 - \eta$ and $2\eta - \tau > 1$, we have $y^{\mathrm{CF}} = 1$. Thus, we have

$$\begin{aligned}
r_{\mathrm{ivd}}(\boldsymbol{x}^{\mathrm{CF}}) &\overset{(b)}{\leq} 1 - \frac{\mathbb{P}(h'(\boldsymbol{x}^{\mathrm{CF}}) \geq \eta)}{\mathbb{P}(h'(\boldsymbol{x}^{\mathrm{CF}}) \geq \eta - \tau)} \\
&\leq 1 - \mathbb{P}(h'(\boldsymbol{x}^{\mathrm{CF}}) \geq \eta) \\
&= 1 - \mathbb{P}(s'(\boldsymbol{x}^{\mathrm{CF}}, 1) \leq 1 - \eta) \\
&\leq 1 - \mathbb{P}(s'(\boldsymbol{x}^{\mathrm{CF}}, y^{\mathrm{CF}}) \leq s_{[k_1^*]}) \\
&\leq \alpha_1 + \hat{\Delta}_F,
\end{aligned}$$

where (b) is from equation 10.

Similarly, if $h(\boldsymbol{x}^{\mathrm{CF}}) > 1 - \eta + \tau$, according to Assumption 2, we have $h'(\boldsymbol{x}^{\mathrm{CF}}) \geq h(\boldsymbol{x}^{\mathrm{CF}}) - \tau > 1 - \eta$, which also causes a contradiction in equation 17. Thus, we have $y^{\mathrm{CF}} = 1$ and $r_{\mathrm{ivd}}(\boldsymbol{x}^{\mathrm{CF}}) \leq \alpha_1 + \hat{\Delta}_F$.

### D.5   Proof of Theorem 10

$$
\begin{aligned}
r_{\mathrm{ivd}}(\boldsymbol{x}^{\mathrm{CF}}) &\leq 1 - \frac{\mathbb{P}(h'(\boldsymbol{x}^{\mathrm{CF}}) \geq \eta)}{\mathbb{P}(h'(\boldsymbol{x}^{\mathrm{CF}}) \geq \eta - \tau)} \\
&= 1 - \frac{\mathbb{P}(s'(\boldsymbol{x}^{\mathrm{CF}}, 1) \leq 1 - \eta)}{\mathbb{P}(s'(\boldsymbol{x}^{\mathrm{CF}}, y^{\mathrm{CF}}) \leq 1 - \eta + \tau)} \\
&\leq 1 - \frac{\mathbb{P}(s'(\boldsymbol{x}^{\mathrm{CF}}, 1) \leq 1 - \eta)}{\mathbb{P}(s'(\boldsymbol{x}^{\mathrm{CF}}, y^{\mathrm{CF}}) \leq s_{[k_2^*]})} \\
&\leq 1 - \frac{\mathbb{P}(s'(\boldsymbol{x}^{\mathrm{CF}}, y^{\mathrm{CF}}) \leq s_{[k_1^*]})}{\mathbb{P}(s'(\boldsymbol{x}^{\mathrm{CF}}, y^{\mathrm{CF}}) \leq s_{[k_2^*]})} \\
&\leq 1 - \frac{1 - \alpha_1 - \hat{\Delta}_F}{1 - \alpha_2 + \hat{\Delta}_E},
\end{aligned}
$$

where $\alpha_2 = 1 - \hat{E}(k_2^*)$ with $k_2^* = \min\{k_2 : s_{[k_2]} \geq 1 - \eta + \tau\}$.

