# OpenReview forum: "Robust Algorithmic Recourse Design Under Model Shifts"
_TMLR — Rejected by TMLR_

### Review · Reviewer_f2Rs · 2024-02-04

**Summary Of Contributions:**

As far as I see, this paper claims to deal with recourse design problems, in particular aiming to ensure robustness to shifts in the data and/or the model. The broad area of algorithmic recourse design with robustness is well-motivated e.g. by Upadhyay et al. (2021), with good discussions of the relevant literature (inc. literature addressing the problem under different terminology).

The problem of recourse is motivated by the desire to suggest modifications on the features that could turn an unfavourable outcome into a favourable one. Plenty of real-world situations in which this problem is relevant, including reasoning about consequences of decisions in high-stakes applications.

An important aspect of the paper under review is combining ideas/techniques from conformal prediction into the algorithmic recourse problem. I think this is interesting and I look forward to reading an upgraded version of the manuscript.

**Audience:**

Yes

**Broader Impact Concerns:**

I think this line of work merits a suitable discussion of broader impact.

**Claims And Evidence:**

No

**Requested Changes:**

**Critical to securing my recommendation:**
- Introduction (Section 1): Previous works talked about a "probability of invalidation" while your work claims "we introduce the concept of the recourse invalidation rate under model shift" which appears to be a probability of invalidation, tailored to model shift? Would be worth to add discussions of the proposed definition in relation to the one used in the related literature.
- Problem formulation (Section 3): An explicit description is missing of the distribution(s) behind the data. Without this, the meaning of probabilities like $\mathbb{P}[Y=1 | \mathbf{X}=\boldsymbol{x}]$ is unclear.
- Definition 1: Could the nonconformity score $s(\boldsymbol{x},y)$ not be written as $s(\boldsymbol{x},y)=(1-y)h(\boldsymbol{x}) + y(1-h(\boldsymbol{x}))$?
- *Remark 2*: I suggest to remove the numbering from this and other remarks in the paper. Numbering these remarks is not adding any value, and at the same time these remark numberings aren't called (cross-referenced) in other parts of the paper. Hence numbering the remarks seems unnecessary, while the visual format is so that from "Definition 1" to "Assumption 3" it looks like the numbering went wrong (skipped one). Then again, better to remove the numbering for remarks.
- Assumption 3: After stating the assumption, and before the following paragraph, some discussion should be inserted about the anticipated role of the assumption, how restrictive it is or not, cases where it is met etc.
- Middle of page 4: Again, can define with a single formula $s'(\boldsymbol{x},y)=(1-y)h'(\boldsymbol{x}) + y(1-h'(\boldsymbol{x}))$.
- Definition 4: Should this not be called "recourse invalidation probability" perhaps adding "under model shift"? It does appear to be defined as a probability (w.r.t. some currently unspecified distribution), as opposed to a rate which evokes an empirical quantity.
- On this note, again since the problem formulation missed declaring distributions, then the meaning of "$\mathbb{P}[f'(\boldsymbol{x}^\mathrm{CF})=0 | f(\boldsymbol{x}^\mathrm{CF})=1]$" is unclear. Note that the distributions could be assumed to be unknown or inaccessible to the learner, but still need to be declared for reasoning about them and for clarity about the meaning of probabilities.
- The paragraph after Definition 4 raised another question: It says that "for a given recourse $\boldsymbol{x}^{\mathrm{CF}}$ [..]" but the definition does not appear to be conditional on $\boldsymbol{x}^{\mathrm{CF}}$, rather appears that the probability assigns mass to a set of $\boldsymbol{x}^{\mathrm{CF}}$. This could be another disadvantage of not having declared the distribution(s) previously in the problem formulation. In any case, this needs clarification.
- Next paragraph: The word "characterize" suggests something stronger than what you do, I think. A legit characterization would be not just deriving some upper bound $r_u(\boldsymbol{x}^{\mathrm{CF}})$ but showing that such bound is the only possible one meeting some criteria. That's what I understand by characterization. You may want to reformulate and write "we aim to derive $r_u(\cdot)$ so that" or something to that effect.
- Another thought: It is not at all obvious that there is a clear advantage in the notation $\boldsymbol{x}^{\mathrm{CF}}$. Would it not suffice to use e.g. $\boldsymbol{x}'$ as the previous literature has done? It seems that $\boldsymbol{x}^{\mathrm{CF}}$ is more cumbersome while not clear what advantages it brings in.
- Section 4: Once again, I am finding trouble understanding ecxactly what the authors have in mind. The opening this section says "for a pre-computed recourse $\boldsymbol{x}^{\mathrm{CF}}$ [..]" and then in Lemma 5 one reads some probabilities... I am wondering what is random in these probabilities, and what is considered as fixed (non-random).
- Without clear specification of the random quantities implied in the probabilities, and their distributions, it is not possible to understand the definitions (Def. 4) and statements of the results (Lemma 5 and other). I am flagging this as a serious flaw that needs to be addressed before this paper can be considered for publication. This is because without the clear unambiguous meaning of these probabilities it is not possible for me (and likewise for readers) to understand even the intended meaning of the definitions/results.
- At this point I decided to submit my review with a note that significant revision is needed.

**Non-critical comments (to help improve the paper, in my view):**
- Abstract: Currently a bit out of focus, need to be sharpened. Try to clearly frame the problem setting, and then briefly highlight this paper's contributions/achievements. See e.g. how Upadhyay et al. (2021) do the abstract.
- Introduction: Ideally also try to illustrate the problem setting with real applications, to improve motivations.
- Format of citations (textual vs parenthetical): It appears that all citations have been set in the textual format (\citet). This format is suitable in some places in the paper, but many other places require the parenthetical format (\citep). Another comment on the format of citations: Not sure why the month needs to be indicated, would it not be sufficient with the Author (Year) format?
- Need to capital-protect titles in the references: It appears that titles in the list of references aren't capital protected currently. This needs to be fixed, so that acronyms and proper names are shown correctly, e.g. EU, GDPR, ADMM, Douglas-Rachford, Bayesian, UCI.
- Capitals for conference names: Similar comments for conference names.

**Strengths And Weaknesses:**

**Strengths:**
- I count as strength the idea of bringing in conformity measures into algorithmic recourse problems.

**Weaknesses:**
- Main weakness is in the execution of the idea: The paper has significant flaws in the writing/presentation. The flaws are significant enough that they may affect the ability of readers to understand the work and/or to verify the arguments.
- Part of the said in the previous bullet affects the writing/narrative, while part is regarding the mathematical framework.
- Also flaws regarding the formatting of references, in particular citation format.

See detailed comments in "Requested Changes" below.

---

> ### Author Response · Authors · 2024-02-23
> **Response to Reviewer f2Rs**
>
> Requested changes:
> 1. Thanks for the comment. We agree with the reviewer that this work builds upon the foundational idea of the probability of invalidation, but with a specific focus on the context of model shifts. In our paper, we define the 'recourse invalidation rate under model shift' to specifically address the dynamic nature of predictive models. In Section 2, under 'user-involved recourse design,' we refer to the work of Pawelczyk et al. (2022) who also discuss a probability of invalidation. However, as we elaborate in that section, Pawelczyk et al.'s focus is on recourse robustness in the context of noisy human responses, where invalidation primarily arises from user actions. In contrast, our work centers on invalidation resulting from model shifts, which is a different and equally critical aspect of recourse robustness.
> 2. Problem formulation (Section 3): Thank you for highlighting the need for clarity in our description of the distribution. To address this, we have made revisions to our manuscript to make them more clear and to prevent any potential misunderstandings. The notation $\mathbb{P}(Y=1 \vert {X}={x})$ in the first paragraph of Section 3 is intended to represent the probability of a positive outcome given the feature vector ${x}$, as derived from the dataset $\mathcal{D}$'s underlying distribution. However, we realized that this notation might lead to confusion later in this section where the probabilities are based on the distribution of the updated classifier $f'$. Thus, we have removed the notation $\mathbb{P}(Y=1 \vert {X}={x})$ from the first paragraph of Section 3 because it is not important and just provides the formula of "probability of a positive outcome given the input data".
> 3. Definition 1: Thanks for your suggestion regarding the expression of the nonconformity score. We agree that the nonconformity score can be simplified and written as $s({x},y)=(1-y)h({x})+y(1-h({x}))$. We have updated our manuscript to include this revised formulation.
> 4. Remark 2: Thank you for your suggestion. We have removed the numbering from all remarks in the document.
> 5. Assumption 3: Thanks for the comment. We have added a discussion after stating this assumption in the revised paper.
> 6. Middle of page 4: Thanks for the suggestion. We have updated our manuscript to include this revised formulation.
> 7. Definition 4: We agree with the reviewer that it is a recourse invalidation probability under model shift just as we have explained in the paragraph that follows this definition. In the revised paper, we have revised Definition 4 (Definition 3 in the revised paper) to include 'under model shift' in the definition.
> 8. Thanks for the comment. The probability $ \mathbb{P}(f'({x}^{\text{CF}})=0 \vert f({x}^{\text{CF}})=1)$ is taken over the distribution of a future classifier $f'$. As shown in Figure 1, following the deployment of the current classifier, future data shifts may occur. In response to these shifts, the model would be retrained using newly collected data, leading to a new classifier $f'$. In practical scenarios, the specifics of future data shifts and the consequent updated model $f'$ are typically unknown. Thus, $f'$ is treated as a random function in our analysis. The randomness in $f'$ comes from possible future data shifts and the associated model adaptations. To address the inherent uncertainty, Assumption 3 (Assumption 2 in the revised paper) constrains the variation in the score function post-shift. Specifically, it constrains the future classifier's score function $h'({x})$ to a small perturbation range around the current score function $h({x})$. We can define the constraint set as $\mathcal{C}_f=\{f': {h({x})} -\tau \leq {h'({x})} \leq {h({x})} +\tau, \forall {x} \in \mathcal{X}\}$. Our theoretical analysis is focused on classifiers $f'$ within this constraint set, i.e. $f' \in \mathcal{C}_f$. We have added this discussion in the revised paper.
> 9. Please see our answer to the above question about $ \mathbb{P}(f'({x}^{\text{CF}})=0 \vert f({x}^{\text{CF}})=1)$.
> 10. Next paragraph: Thank you for pointing out the potential misinterpretation of the term 'characterize' in our manuscript. We have rephrased the sentence in the revised paper accordingly.

---

> ### Author Response · Authors · 2024-02-23
> **Continued response**
>
> Requested changes:
>
> 11. Another thought: Thank you for your comment regarding the notation ${x}^{\text{CF}}$. We understand your concern about the potential cumbersomeness of this notation. However, we have chosen to use ${x}^{\text{CF}}$ in line with established conventions in the literature on algorithmic recourse. In this domain, ${x}^{\text{CF}}$ is commonly used to denote the counterfactual explanation, or the recourse. While we agree with the reviewer that other notations like ${x}'$ are simpler, they might not convey the specific meaning of counterfactual explanations.
>
> 12. Section 4: Thanks for the comment. In Section 4, the counterfactual instant ${x}^{\text{CF}}$ is considered fixed and given. The classifier $f'$ is treated as a random function, reflecting the inherent uncertainty in future data shifts and the consequent changes in the classifier. Consequently, the nonconformity score $s'$, which is dependent on $f'$, is also random. In the revised paper, we have clarified that the primary source of randomness in the analysis is with $f'$.
>
> Non-critical comments:
> 1. Abstract: We appreciate your feedback on sharpening the focus of our abstract. Inspired by the structure used by Upadhyay et al. (2021), we have revised the abstract to more clearly define the problem setting and highlight the key contributions of our paper.
> 2. Introduction: Thank you for the suggestion to illustrate the problem setting with real-world applications. We have incorporated a mortgage application example in the first paragraph of the introduction. This example provides relevant context and strengthens the motivation of our paper. Additionally, to illustrate the cost-robustness trade-off, we have utilized this loan application scenario in Figure 2.
> 3. Format of Citations: Thanks for the comment. We have revised the citations throughout the paper, ensuring the appropriate use of textual and parenthetical formats as required. We have also omitted month details in the citation.
> 4. Need to capital-protect titles in the references: We acknowledge the importance of correctly capitalizing titles in the references. The titles have been updated.
> 5. Capitals for conference names: Thank you for the comment. We have revised the reference section to ensure that all conference names are accurately capitalized.

---

> ### Comment · Reviewer_f2Rs · 2024-03-11
> **acknowledgement of authors' response**
>
> I thank the authors for taking into account my feedback.
>
> Thanks for responding to my comments and updating the manuscript.
>
> One further comment: In Line 14 of the introduction, it appears that "(AR)" is unexplained. Also it is not clear what's the purpose of inserting this acronym at this place. I thought may be the intention was to attribute to Ustun et al. (2019) but that paper does not seem to use the acronym "AR" so that might not be the intended. If "AR" is meant to stand for "actionable recourse" then it would be better to move "(AR)" to after "list of actionable recourses" to imply this meaning. It appears there is related literature that uses "AR" in this sense (actionable recourse). Other acronyms later like ROAR suggest that AR could stand for "algorithmic recourse" so again better to clarify explicitly. Or to delete "(AR)" if the acronym is not used later in the paper, as appears to be the case.

---

> > ### Author Response · Authors · 2024-03-12
> > **Response to reviewer**
> >
> > Further comment:
> >
> > Thank you for highlighting the ambiguity about the acronym "AR" in Line 14 of the introduction. "AR" was intended to refer to "actionable recourse".  However, recognizing that its inclusion at this point in the manuscript does not add substantial value and following the reviewer's suggestion, we have removed it from the paper.

---

### Review · Reviewer_nqYV · 2024-02-07

**Summary Of Contributions:**

The paper considers the robustness of the recommended recourse action in the presence of shifts in data distribution and model itself. In particular, under certain conditions/assumptions, the paper aims to provide a theoretical upperbound of the recourse invalidation rate, and furthermore, to propose a framework to endow users with the ability to choose between different tradeoffs between recourse cost and robustness. Theoretical derivations and empirical experiments are presented.

**Audience:**

Yes

**Broader Impact Concerns:**

The paper focuses on the relation between recourse cost and robustness, which does not seem to raise ethical concerns.

**Claims And Evidence:**

Yes

**Requested Changes:**

It would be very helpful is points/questions in Section "Strengths And Weaknesses" can be clearly addressed.

**Strengths And Weaknesses:**

The strength of the paper comes from the relatively careful review and discussion of the related works, and the clear presentation of how the current work fits in the related literature. The paper is overall not hard to follow, and the presentation is overall clear.

The weakness of the paper comes from the gap between the ambitious goal ("theoretical upperbound of the recourse invalidation rates of _any_ counterfactual plan and _any_ prediction model without [any] distributional assumption about feature space") and the actual theoretical analysis. In particular, clarifications on following points/questions would be helpful:

**Q1** how to parse (in terms of practical applicability) of Assumption 3?

Assumption 3 directly address the difference between the prediction function before and after shifts. From a purely technical point of view, Assumption 3 seems natural (to a certain extent, too convenient). While there is indeed no distributional assumption about feature space, Assumption 3 essentially encapsulates different assumptions about how the shift shapes the feature distribution (i.e., feature space assumptions) and/or model behavior. In other words, Assumption 3 is too general to be interpretable in practical scenarios, making it a convenient condition only for technical purposes.

**Q2** regarding the other side of recourse invalidation rate

While I understand that in Definition 4 (and also Lemma 5), the recourse invalidation rate only pertains to the situation where the recourse action works before shifts but no longer works after shifts. I am wondering if the framework can also capture the other side of the story, i.e., certain action may not suffice before shifts, but can result in favorable results after shifts. This does not violate any specified assumptions in the paper, and seems to be a reasonable possibility, now that the paper is aiming for a rather general bound with minimal assumptions.

---

> ### Author Response · Authors · 2024-02-23
> **Response to Reviewer nqYV**
>
> Q1: Thank you for raising the question regarding Assumption 3 (Assumption 2 in the revised paper) and its practical applicability. In our theoretical framework, Assumption 3 (Assumption 2 in the revised paper) serves as a guideline rather than a strict requirement for all practical applications. Specifically, for a counterfactual example ${x}^{\text{CF}}$, it is not necessary that the difference between the scoring functions before and after a model shift is smaller than $\tau$ for every sample. Instead, the more practical condition we use is: $$\mathbb{P}(h'({x}^\text{CF})\geq \eta-\tau \vert (h'({x}^\text{CF})\geq \eta)=1.$$
> This condition reflects a more realistic scenario where the bound $\tau$ serves as a probabilistic rather than an absolute constraint.
>
> Our decision to introduce Assumption 3 (Assumption 2 in the revised paper) is based on common occurrences in dynamic decision systems, such as model updates or changes in data distribution. We recognize that these shifts are typically not extreme, hence the theoretical assumption that perturbations are bounded by $\tau$.
>
> While our theoretical model assumes bounded perturbations for simplicity and tractability, we acknowledge the complexities of real-world data and model shifts. In light of this, one of our key future research directions is to relax this assumption, allowing for a broader range of perturbations. By relaxing this assumption, we aim to further bridge the gap between theoretical robustness and the practical variability encountered in real-world applications.
>
> Q2: We appreciate your insightful observation regarding the possibility of certain recourses not being effective before model shifts, but becoming favorable after shift. This represents an interesting aspect of model dynamics that could be relevant in some scenarios. We believe that such recourse action might align with the anticipated trend of model shifts. However, such an action might only be temporarily effective due to particular model configurations during a certain period and may not necessarily reflect a consistent, long-term trend. Additionally, considering that the timing of a user's implementation of recourse actions is typically uncertain, a recourse's effectiveness during a specific period may not have substantial practical significance.
>
> In this paper, our primary goal is to provide users with actionable recommendations that can reliably alter unfavorable outcomes in current and also futue decision-making systems. The focus on the recourse invalidation rate is meant to evaluate the robustness of these recommendations in the face of future model shifts. While the scenario you described is indeed a valid consideration, incorporating it into our current analysis could potentially divert attention from our main objective. Nevertheless, this aspect could certainly be an interesting direction for future research.

---

### Review · Reviewer_eEUV · 2024-02-10

**Summary Of Contributions:**

The paper proposes an uncertainty quantification strategy for bounding the so-called "recourse invalidation rate," which is the rate at which a point that is identified as favorable for the current classifier will be negatively classified in the future. The paper also proposes a way of generating recourses that satisfy the user's tolerance for the invalidation rate. The theory is accompanied by simulations on several datasets.

**Audience:**

Yes

**Broader Impact Concerns:**

It would be good to include a discussion of potential failure modes of the proposal (e.g. what happens if we have a wrong estimate of tau, etc).

**Claims And Evidence:**

No

**Requested Changes:**

Please look at my comments above. The theory needs to be made rigorous before I can give a well-informed opinion. Right not the setup is ill-defined.

**Strengths And Weaknesses:**

This is a timely topic and a contribution along the lines of what this paper proposes is worthwhile. While reading the intro I was excited about the potential contribution of this paper. However, after looking at the technical details, in its current form this paper does not make mathematical sense to me and I can't give a well-informed and favorable review until the theory is formalized.

For example, in Definition 4, what is the probability taken over? x^{CF} is a fixed argument to the function, and f and f' are talked about as being fixed in the preceding paragraphs (my understanding was that the training data was conditioned on throughout). If f' is treated as random function, what data is used to obtain f'? Similarly, after Lemma 5 it says that s'(x^{CF}, 1) "quantifies the degree to which the counterfactual sample (x^{CF}, 1) conforms to the calibration samples in D_{calib}." How is this possible when s'(x^{CF}, 1) is a fixed nonconformity score with a fixed data point as the argument? There should be no dependence of s'(x^{CF}, 1) on the calibration data. After Lemma 7, when defining L-hat and U-hat in point 3, what if there is no s' in S_{train} within tau of s? Then U-hat(s) would be equal to 0, which shouldn't be allowed.

Some other comments:
- Conformal prediction dates back to Vovk's work. Please attribute it correctly.
- In the setup of Section 3, there is no point in introducing the function g(t) if right after you're going to set it to be 1{t>= eta}. Defining an abstract g introduces unnecessary complexity.
- It would be good to clarify why the user-specified invalidation rate has an index t.
- I don't understand the criterion according to which the entries in Table 3 and Table 4 are bolded.

---

> ### Author Response · Authors · 2024-02-23
> **Response to Reviewer eEUV**
>
> In Definition 4 (Definition 3 in the revised paper), the probability is taken over the distribution of $f'$. We agree with the reviewer that ${x}^{\text{CF}}$ is a fixed argument. For $f'$, it is trained based on potentially updated data, which is unspecified in the setup of this paper. As shown in Figure 1, following the deployment of the current classifier, future data shifts may occur. In response to these shifts, the model would be retrained using newly collected data, leading to a new classifier $f'$. In practical scenarios, the specifics of future data shifts and the consequent updated model $f'$ are typically unknown. Thus, $f'$ is treated as a random function in our analysis. The randomness in $f'$ comes from possible future data shifts and the associated model adaptations. This inherent uncertainty about the future classifier $f'$ is a key aspect of our analysis.
>
> To address this uncertainty, we introduce Assumption 3 (Assumption 2 in the revised paper), which limits the variation in the model's score function post-shift. Specifically, it constrains the future classifier's score function $h'({x})$ to a small perturbation range around the current score function $h({x})$. In particular, we define the constraint set as $\mathcal{C}_f=\{f': {h({x})} -\tau \leq {h'({x})} \leq {h({x})} +\tau, \forall {x} \in \mathcal{X}\}$. The theoretical analysis in Section 4 is for $f' \in \mathcal{C}_f$.
>
> Moreover, since $f'$ is treated as a random function, the noncomformity score $s'$ is also random. Consequently, the value of $s'({x}^{\text{CF}},1)$ is not fixed and depends on the realization of $f'$ within $\mathcal{C}_f$.
>
> For the comment regarding the definition of $\hat{L}$ and $\hat{U}$, we acknowledge the concern about the possibility of no $s'$ in $S_{\text{train}}$ within $\tau$ of $s$. However, such a scenario will not arise due to the constriants imposed by Assumption 3 (Assumption 2 in the revised paper). The proof of Lemma 7 (Lemma 6 in the revised paper), particularly an intermediate result, ensures that for any pair $({x},y)$, $s({x},y)-\tau \leq s'({x},y) \leq s({x},y)+\tau$. In particular, under Assumption 3 (Assumption 2 in the revised paper), we have $ {h({x})} -\tau \leq {h'({x})} \leq {h({x})} +\tau, \forall {x} \in \mathcal{X}$, which implies
> \begin{eqnarray}
> 	{s({x},y=0)} -\tau \leq {s'({x},y=0)} \leq {s({x},y=0)} +\tau, \nonumber
> \end{eqnarray}
> as well as
> \begin{eqnarray}
> 	{s({x},y=1)} -\tau=1- {h({x})} -\tau \leq 1-{h'({x})}={s'({x},y=1)} \leq 1- {h({x})} +\tau = {s({x},y=1)} +\tau.  \nonumber
> \end{eqnarray}
> As the result, we have
> \begin{eqnarray}
> 	s({x},y)-\tau \leq s'({x},y) \leq s({x},y)+\tau. \nonumber
> \end{eqnarray}
> Therefore, it is guaranteed that there will always be scores in $S_{\text{train}}$ within $\tau$ of any given $s$, ensuring that $\hat{U}(s)$ is never zero.
>
> Other comments:
> 1. Thanks for the comment. We have added Vovk's work in the revised paper.
> 2. We want to use $g$ to denote the activation function as that is offten used in classification models. Similar notations have also been used in (Pawelczyk et al., 2022).
> 3. In this paper, users have the flexibility to specify multiple invalidation rates. This is illustrated in Section 5 of our paper, where we introduce a series of user-specified invalidation rates, denoted as $\gamma_1,\gamma_2,\cdots, \gamma_m$. Here, $\gamma_t$ refers to the $t$-th invalidation rate in this series.
> 4. In Tables 3 and 4, the bolded entries highlight the algorithms that best align with the key objectives of algorithmic recourse in practical scenarios. Our primary goals are twofold: ensuring the validity of generated recourses both before and after model shifts, and minimizing the implementation costs to enhance feasibility for users. Each table contains three key columns evaluating each algorithm's performance: the invalidation rate before the shift, the invalidation rate after the shift, and the average implementation cost. The bolded entries denote the algorithms that perform exceptionally well according to these specific metrics. Specifically, algorithms with lower invalidation rates (both before and after the shift) and a lower average cost are highlighted in bold.

---

### Decision · Action_Editor_zxr8 · 2024-03-26

**Recommendation:** Reject

**Comment:**

The robust algorithmic recourse design is a well-motivated topic (Reviewers eEUV and f2Rs). However, the execution of the paper is not rigorous (eEUV, nqYV and f2Rs). Particularly, as the reviewers pointed out:

- Assumption 2 is too general and "the gap between the ambitious goal and the presented analysis is not resolved" in the revision (Reviewer nqYV).

- "The paper has significant flaws in the writing/presentation." (Reviewer f2Rs and nqYV).

Minor:
For the equivalent between Eq (1) and (2) claim, why f' is defined as f - 1, while the authors claim f' is random?

After the discussion, the majority of the reviewers voted for rejection at this moment. I sincerely hope the authors can further improve the clarity and solve the interesting problem in a rigorous way.

**Audience:**

Yes,  as the reviewer eEUV and f2Rs pointed, the topic for algorithmic recourse with model shift is an interesting topic for the community.

**Claims And Evidence:**

This paper investigated the algorithmic recourse under model shift. The algorithm is derived based on a conservative assumption that for any data, the model shift is uniformly upper and lower bounded.

The authors evaluated the performances of the proposed algorithm on empirical data. However, the benefits of the proposed algorithm is not clearly justified.

**Resubmission Of Major Revision:**

The authors may consider submitting a major revision at a later time.